# MetaGL: Evaluation-Free Selection of Graph Learning Models via Meta-Learning

**Namyong Park**
Carnegie Mellon University
namyongp@cs.cmu.edu

**Ryan Rossi**
Adobe Research
ryrossi@adobe.com

**Nesreen Ahmed**
Intel Labs
nesreen.k.ahmed@intel.com

**Christos Faloutsos**
Carnegie Mellon University
christos@cs.cmu.edu

## Abstract

Given a graph learning task, such as link prediction, on a new graph, how can we select the best method as well as its hyperparameters (collectively called a model) without having to train or evaluate any model on the new graph? Model selection for graph learning has been largely ad hoc. A typical approach has been to apply popular methods to new datasets, but this is often suboptimal. On the other hand, systematically comparing models on the new graph quickly becomes too costly, or even impractical. In this work, we develop the *first meta-learning approach for evaluation-free graph learning model selection*, called METAGL, which utilizes the prior performances of existing methods on various benchmark graph datasets to automatically select an effective model for the new graph, *without* any model training or evaluations. To quantify similarities across a wide variety of graphs, we introduce specialized meta-graph features that capture the structural characteristics of a graph. Then we design G-M network, which represents the relations among graphs and models, and develop a graph-based meta-learner operating on this G-M network, which estimates the relevance of each model to different graphs. Extensive experiments show that using METAGL to select a model for the new graph greatly outperforms several existing meta-learning techniques tailed for graph learning model selection (up to *47%* better), while being extremely fast at test time (∼*1 sec*).

## 1 Introduction

Given a graph learning (GL) task, such as link prediction, for a new graph dataset, how can we select the best method as well as its hyperparameters (HPs) (collectively called a model) *without* performing any model training or evaluations on the new graph? GL has received increasing attention recently (Zhang et al., 2022), achieving successes across various applications, *e.g.*, recommendation and ranking (Fan et al., 2019; Park et al., 2020), traffic forecasting (Jiang & Luo, 2021), bioinformatics (Su et al., 2020), and question answering (Park et al., 2022). However, as GL methods continue to be developed, it becomes increasingly difficult to determine which model to use for the given graph.

Model selection (*i.e.*, selecting a method and its configuration such as HPs) for graph learning has been largely ad hoc to date. A typical approach, called "no model selection", is to simply apply popular methods to new graphs, often with the default HP values. However, it is well known that there is no universal learning algorithm that performs best on *all* problem instances (Wolpert & Macready, 1997), and such consistent model selection is often suboptimal. At the other extreme lies "naive model selection" (Fig. 1b), where all candidate models are trained on the new graph, evaluated on a hold-out validation graph, and then the best performing model for the new graph is selected. This approach is very costly as all candidate models are trained when a new graph arrives. Recent methods on neural architecture search (NAS) and hyperparameter optimization (HPO) of GL methods, which we review in Section 3, adopt smarter and more efficient strategies, such as Bayesian optimization (Snoek et al., 2012; Tu et al., 2019), which carefully choose a relatively small number of HP settings to evaluate. However, they still need to evaluate multiple configurations of each GL method on the new graph.

Evaluation-free model selection is yet another paradigm, which aims to tackle the limitations of the above approaches by attempting to simultaneously achieve the speed of no model selection and the accuracy of exhaustive model selection. Recently, a seminal work by Zhao et al. (2021) proposed a technique for outlier detection (OD) model selection, which carries over the observed performance of

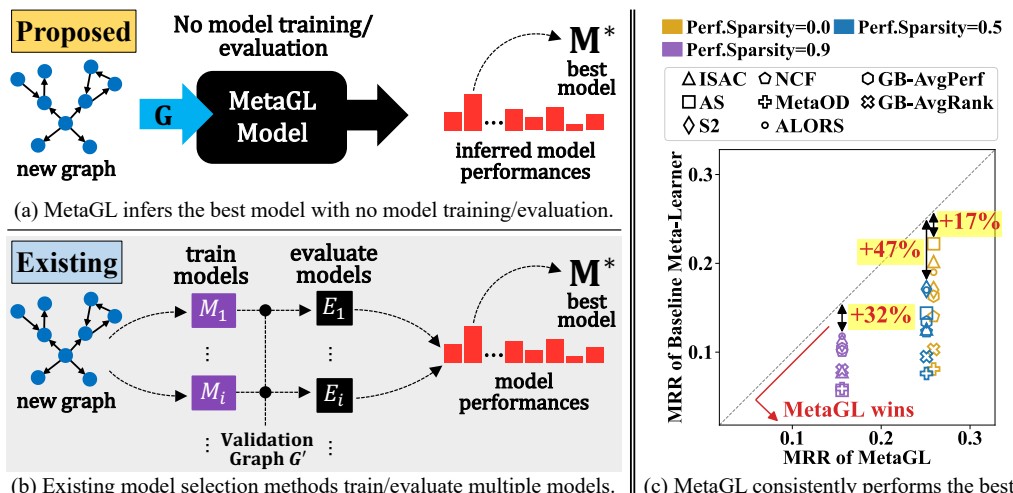

(a) MetaGL infers the best model with no model training/evaluation.

(b) Existing model selection methods train/evaluate multiple models.

(c) MetaGL consistently performs the best.

Figure 1: (a) Given an unseen graph $G$ and a large space $\mathcal{M}$ of models to search over, METAGL efficiently infers the best model $M^* \in \mathcal{M}$ without having to train a single model from $\mathcal{M}$ on the new graph $G$. (b) Existing approaches, by contrast, need to train and evaluate multiple models $M_i \in \mathcal{M}$ to be able to select the best one. (c) Given observed performance with varying sparsity levels, METAGL consistently outperforms existing meta-learners, with up to 47% better model selection performance.

OD methods on benchmark datasets for selecting OD methods. However, it does not address the unique challenges of GL model selection, and cannot be directly used to solve the problem. Inspired by this work, we systematically tackle the model selection problem for graph learning, especially link prediction. We choose link prediction as it is a key task for graph-structured data: It has many applications (*e.g.*, recommendation, knowledge graph reasoning, and entity resolution), and several inference and learning tasks can be cast as a link prediction problem (*e.g.*, Fadaee & Haeri (2019)). In this work, we develop METAGL, *the first meta-learning framework for evaluation-free selection of graph learning models*, which finds an effective GL model to employ for a new graph *without* training or evaluating any GL model on the new graph, as Figure 1a illustrates. METAGL satisfies all of desirable features for GL model selection listed in Table 1, while no existing paradigms satisfy all.

The high-level idea of meta-learning based model selection is to estimate a candidate model's performance on the new graph based on its observed performances on *similar* graphs. Our meta-learning problem for graph data presents a unique challenge of how to model graph similarities, and what characteristic features (*i.e.*, meta-features) of a graph to consider. Note that this step is often not needed for traditional meta-learning problems on non-graph data, as features for non-graph objects (*e.g.*, location, age of users) are often readily available. Also, the high complexity and irregularity of graphs (*e.g.*, different number of nodes and edges, and widely varying connectivity patterns among different graphs) makes the task even more challenging. To handle these challenges, we design specialized *meta-graph features* that can characterize major structural properties of real-world graphs.

Then to estimate the performance of a candidate model on a given graph, METAGL learns to embed models and graphs in the shared latent space such that their embeddings reflect the graph-to-model affinity. Specifically, we design a multi-relational graph called *G-M network*, which captures multiple types of relations among models and graphs, and develop a meta-learner operating on this G-M network, based on an attentive graph neural network that is optimized to leverage meta-graph features and prior model performance into producing model and graph embeddings that can be effectively used to estimate the best performing model for the new graph. METAGL greatly outperforms existing meta-learners in GL model selection (Fig. 1c). In sum, the key contributions of this work are as follows.

- **Problem Formulation.** We formulate the problem of selecting effective GL models in an evaluation-free manner (*i.e.*, without ever having to train/evaluate any model on the new graph), To the best of our knowledge, we are the first to study this important problem.
- **Meta-Learning Framework and Features.** We propose METAGL, the first meta-learning framework for evaluation-free GL model selection. For meta-learning on various graphs, we design meta-graph features that quantify graph similarities by capturing structural characteristics of a graph.
- **Effectiveness.** Using METAGL for GL model selection greatly outperforms existing meta-learning techniques (up to *47%* better, Fig. 1c), with negligible runtime overhead at test time (∼*1 sec*).

**Benchmark Data/Code:** To facilitate further research on this important new problem, we release code and data at `https://github.com/NamyongPark/MetaGL`, including performances of 400+ models on 300+ graphs, and 300+ meta-graph features.

Table 1: The proposed METAGL wins on features in comparison to existing graph learning (GL) model selection (MS) paradigms, all of which fail to satisfy some of desirable properties for GL MS.

| GL Model Selection (MS) Paradigms / Desiderata for Graph Learning (GL) MS | No model selection | Naive model selection (Fig. 1b) | Graph HPO/NAS (e.g., AutoNE, AGNN; See Sec 3 and Fig. 1b) | METAGL (Ours, Fig. 1a) |
|---|---|---|---|---|
| Evaluation-free GL model selection | ✓ | | | ✓ |
| Capable of MS from among multiple GL algorithms | | ✓ | | ✓ |
| Capitalizing on graph similarities for GL MS | | | | ✓ |
| Estimating model performance based on past observations | | | ✓ | ✓ |

## 2 PROBLEM FORMULATION

Given a new unseen graph, our goal is to select the best model from a heterogeneous set of graph learning models, without requiring any model evaluations and user intervention. In comparison to traditional meta-learning problems where a model denotes a single method and its hyperparameters, a model in the graph meta-learning problem is more broadly defined to be

$$\text{model } M = \{(\text{graph embedding method}, \text{hyperparameters}), (\text{predictor}, \text{hyperparameters})\}, \quad (1)$$

as graph learning tasks usually involve two steps: (1) embedding the graph using a graph representation learning method, and (2) providing node embeddings to the predictor of a downstream task like link prediction. Both steps require learning a method with specific hyperparameters. Thus, there can be many models with the same embedding method and predictor, which have different hyperparameters. Also, the set $\mathcal{M}$ of models may contain many different graph representation learning methods (*e.g.*, node2vec (Grover & Leskovec, 2016), GraphSAGE (Hamilton et al., 2017), DeepGL (Rossi et al., 2020) to name a few), as well as multiple task-specific predictors, making $\mathcal{M}$ *heterogeneous*.

Given a training meta-corpus of $n$ graphs $\mathcal{G} = \{G_i\}_{i=1}^n$, and $m$ models $\mathcal{M} = \{M_j\}_{j=1}^m$ for GL tasks, we derive performance matrix $\mathbf{P} \in \mathbb{R}^{n \times m}$ where $P_{ij}$ is the performance (*e.g.*, accuracy) of model $j$ on graph $i$. Our meta-learning problem for evaluation-free GL model selection is defined as follows.

**Problem 1** (Evaluation-Free Graph Learning Model Selection).

**Given**
- an unseen test graph $G_{\text{test}} \notin \mathcal{G}$, and
- a potentially sparse performance matrix $\mathbf{P} \in \mathbb{R}^{n \times m}$ of $m$ *heterogeneous* graph learning models $\mathcal{M} = \{M_1, \ldots, M_m\}$ on $n$ graphs $\mathcal{G} = \{G_1, \ldots, G_n\}$,

**Select**
- the best model $M^* \in \mathcal{M}$ to employ on $G_{\text{test}}$ *without* evaluating any model in $\mathcal{M}$ on $G_{\text{test}}$.

## 3 RELATED WORK

A majority of works on GL focus on developing new algorithms for certain graph tasks and applications (Xia et al., 2021; Zhang et al., 2022). In comparison, there exist relatively few recent works that address the GL model selection problem (Zhang et al., 2021). They mainly focus on neural architecture search (NAS) and hyperparameter optimization (HPO) for GL models, especially graph neural networks (GNNs). Toward efficient and effective NAS and HPO in GL, they investigated several approaches, such as Bayesian optimization (AutoNE by Tu et al. (2019)), reinforcement learning (GraphNAS by Gao et al. (2020), AGNN by Zhou et al. (2019), Policy-GNN by Lai et al. (2020)), hypernets (ST-GCN by Zhu et al. (2021)), and evolutionary algorithms (Bu et al. (2021)), as well as techniques like subgraph sampling (AutoNE by Tu et al. (2019)), graph coarsening (JITuNE by Guo et al. (2021)), and hierarchical evaluation (HESGA by Yuan et al. (2021)). However, as summarized in Table 1, these methods cannot perform evaluation-free GL model selection (Problem 1), since they need to evaluate multiple configurations of each GL method on the new graph for model selection. Further, they are limited to finding the best configuration of a single algorithm, and thus cannot select a model from a heterogeneous model set $\mathcal{M}$ with various GL models, as Problem 1 requests. An earlier work on GNN design space (You et al., 2020) is somewhat relevant, as it proposes an approach to quantify graph similarities, which can be used to find an observed graph similar to the test graph, and select a model that performed best on it. However, their approach evaluates a set of anchor models on all graphs, and computes similarities between two graphs based on anchor models' performance on them. As it needs to run anchor models on the new graph, it is inapplicable to Problem 1. For the first time, METAGL enables evaluation-free model selection from a heterogeneous set of GL models.

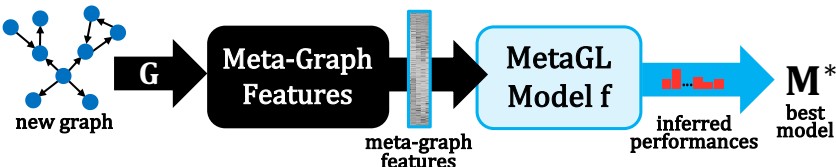

Figure 2: Given a new graph $G$, METAGL extracts meta-graph features, and applies a meta-learned model to them, which efficiently infers the best model $M^* \in \mathcal{M}$ for $G$, with no model evaluation.

## 4 FRAMEWORK

In this section, we present METAGL, our meta-learning based framework that solves Problem 1 by leveraging prior performances of existing methods. METAGL consists of two phases: (1) *offline meta-training* phase (Section 4.1) that trains a meta-learner using observed graphs $\mathcal{G}$ and model performances $\mathbf{P}$, and (2) *online model prediction* phase (Section 4.2), which selects the best model for the new graph. A summary of notations used in this work is provided in Table 3 in the Appendix.

### 4.1 OFFLINE META-TRAINING

Meta-learning leverages prior experience from related learning tasks to do a better job on the new task. When the new task is similar to some historical learning tasks, then the knowledge from those similar tasks can be transferred and applied to the new task. Thus effectively capturing the similarity between an input task and observed ones is fundamentally important for successful meta-learning. In meta-learning, the similarity between learning tasks is modeled using *meta-features*, *i.e.*, characteristic features of the learning task that can be used to quantify the task similarity.

**Meta-Graph Features.** Given the graph learning model selection problem (where new graphs correspond to new learning tasks), METAGL captures the graph similarity by extracting *meta-graph features* such that they reflect the structural characteristics of a graph. Notably, since graphs have irregular structure, with different number of nodes and edges, METAGL designs meta-graph features to be of the same size for any arbitrary graph such that they can be easily compared using meta-graph features. We use the symbol $\mathbf{m} \in \mathbb{R}^d$ to denote the fixed-size meta-graph feature vector of graph $G$. We defer the details of how METAGL computes $\mathbf{m}$ to Section 4.3.

**Model Performance Estimation.** To estimate how well a model would perform on a given graph, METAGL represent models and graphs in the latent $k$-dimensional space, and captures the graph-to-model affinity using the dot product similarity between the two representations $\mathbf{h}_{G_i}$ and $\mathbf{h}_{M_j}$ of the $i$-th graph $G_i$ and $j$-th model $M_j$, respectively, such that $p_{ij} \approx \langle \mathbf{h}_{G_i}, \mathbf{h}_{M_j} \rangle$ where $p_{ij}$ is the performance of model $M_j$ on graph $G_i$. Then to obtain the latent representation $\mathbf{h}$, we design a learnable function $f(\cdot)$ that takes in relevant information on models and graphs from the meta-graph features $\mathbf{m}$ and the prior knowledge (*i.e.*, model performances $\mathbf{P}$ and observed graphs $\mathcal{G}$). Below in this section, we focus on the inputs to the function $f(\cdot)$, and defer the details of $f(\cdot)$ to Section 4.4.

We first factorize performance matrix $\mathbf{P}$ into latent graph factors $\mathbf{U} \in \mathbb{R}^{n \times k}$ and model factors $\mathbf{V} \in \mathbb{R}^{m \times k}$, and take the model factor $\mathbf{V}_j \in \mathbb{R}^k$ (the $j$-th row of $\mathbf{V}$) as the input representation of model $M_j$. Then, METAGL obtains the latent embedding $\mathbf{h}_{M_j}$ of model $M_j$ by $\mathbf{h}_{M_j} = f(\mathbf{V}_j)$. For graphs, more information is available since we have both meta-graph features $\mathbf{m}$ and meta-train graph factors $\mathbf{U}$. However, while we have the same number of models during training and inference, we observe new graphs during inference, and thus cannot obtain the graph factor $\mathbf{U}_{\text{test}}$ for the test graph as for the train graphs since matrix factorization (MF) is transductive by construction (*i.e.*, existing models' performance on the test graph is needed to get latent factors for the test graph directly via MF). To handle this issue, we learn an estimator $\phi : \mathbb{R}^d \mapsto \mathbb{R}^k$ that maps the meta-graph features $\mathbf{m}$ into the latent factors of meta-train graphs obtained via MF above (*i.e.*, for graph $G_i$ with $\mathbf{m}$, $\phi(\mathbf{m}) = \hat{\mathbf{U}}_i \approx \mathbf{U}_i$) and use this estimated graph factor. We then combine both inputs ($[\mathbf{m}; \phi(\mathbf{m})] \in \mathbb{R}^{d+k}$), and apply linear transformation to make the input representation of graph $G_i$ to be of the same size as that of model $M_j$, obtaining the latent embedding of graph $G_i$ to be $\mathbf{h}_{G_i} = f(\mathbf{W}[\mathbf{m}; \phi(\mathbf{m})])$ where $\mathbf{W} \in \mathbb{R}^{k \times (d+k)}$ is a weight matrix. Thus in METAGL, the performance $p_{ij}$ of model $M_j$ on graph $G_i$ with meta-graph features $\mathbf{m}$ is estimated as

$$p_{ij} \approx \hat{p}_{ij} = \langle f(\mathbf{W}[\mathbf{m}; \phi(\mathbf{m})]), f(\mathbf{V}_j) \rangle. \tag{2}$$

**Meta-Learning Objective.** For tasks where the goal is to estimate real values, such as accuracy, the mean squared error (MSE) is a typical choice for the loss function. While MSE is easy to optimize and

effective for regression, it does not directly take the ranking quality into account. By contrast, in our problem setup, accurately ranking models for each graph dataset is more important than estimating the performance itself, which makes MSE a suboptimal choice. In particular, Problem 1 focuses on finding the model with the best performance on the given graph. Therefore, we consider rank-based learning objectives, and among them, we adapt the top-1 probability to the proposed Problem 1 as follows. Let $\widehat{\mathbf{P}}_i \in \mathbb{R}^m$ be the $i$-th row of $\widehat{\mathbf{P}}$ (*i.e.*, estimated performance of all $m$ models on graph $G_i$). Given $\widehat{\mathbf{P}}_i$, the top-1 probability $p_{\text{top1}}^{\widehat{\mathbf{P}}_i}(j)$ of $j$-th model $M_j$ in the model set $\boldsymbol{\mathcal{M}}$ is defined to be

$$p_{\text{top1}}^{\widehat{\mathbf{P}}_i}(j) = \frac{\pi(\widehat{p}_{ij})}{\sum_{k=1}^m \pi(\widehat{p}_{ik})} = \frac{\exp(\widehat{p}_{ij})}{\sum_{k=1}^m \exp(\widehat{p}_{ik})} \tag{3}$$

where $\pi(\cdot)$ is an increasing, strictly positive function, which we define to be an exponential function.

**Theorem 1** (Cao et al. (2007)). *Given the performance* $\widehat{\mathbf{P}}_i$ *of all models on graph* $G_i$, $p_{\text{top1}}^{\widehat{\mathbf{P}}_i}(j)$ *represents the probability of model* $M_j$ *to be ranked at the top of the list (i.e., all models in* $\boldsymbol{\mathcal{M}}$*). Top-1 probabilities* $p_{\text{top1}}^{\widehat{\mathbf{P}}_i}(j)$ *for all* $j = 1, \ldots, m$ *form a probability distribution over* $m$ *models.*

Based on Theorem 1, we obtain two probability distributions by applying top-1 probability to the true performance $\mathbf{P}_i$ and estimated performance $\widehat{\mathbf{P}}_i$ of $m$ models, and optimize METAGL such that the distance between the two resulting distributions gets decreased. Using the cross entropy as the distance metric, we obtain the following loss over all $n$ meta-train graphs $\boldsymbol{\mathcal{G}}$:

$$L(\mathbf{P}, \widehat{\mathbf{P}}) = -\sum_{i=1}^n \sum_{j=1}^m p_{\text{top1}}^{\mathbf{P}_i}(j) \log\left(p_{\text{top1}}^{\widehat{\mathbf{P}}_i}(j)\right) \tag{4}$$

When $\mathbf{P}$ is sparse, meta-training can be performed via slightly modified Eqs. (3) and (4) in App. G.2.

## 4.2 ONLINE MODEL PREDICTION

In the meta-training phase, METAGL learns estimators $f(\cdot)$ and $\phi(\cdot)$, as well as weight matrix $\mathbf{W}$ and latent model factors $\mathbf{V}$. Given a new graph $G_{\text{test}}$, METAGL first computes the meta-graph features $\mathbf{m}_{\text{test}} \in \mathbb{R}^d$ as we discuss in Section 4.3. Then $\mathbf{m}_{\text{test}}$ is regressed to obtain the (approximate) latent graph factors $\widehat{\mathbf{U}}_{\text{test}} = \phi(\mathbf{m}_{\text{test}}) \in \mathbb{R}^k$. Recall that the model factors $\mathbf{V}$ learned in the meta-training stage can be directly used for model prediction. Then model $M_j$'s performance on test graph $G_{\text{test}}$ can be estimated by applying Equation (2) with $\mathbf{m}_{\text{test}}$ and $\phi(\mathbf{m}_{\text{test}})$. Finally, the model that has the highest estimated performance is selected by METAGL as the best model $M^*$, *i.e.*,

$$M^* \leftarrow \underset{M_j \in \boldsymbol{\mathcal{M}}}{\arg\max} \ \left\langle f(\mathbf{W}[\mathbf{m}_{\text{test}}; \phi(\mathbf{m}_{\text{test}})]), f(\mathbf{V}_j) \right\rangle \tag{5}$$

Note that model selection using Equation (5) depends only on the meta-graph features $\mathbf{m}_{\text{test}}$ of the test graph and other pretrained estimators and latent factors that METAGL learned in the meta-training phase. As no model training or evaluation is involved, model prediction by METAGL is much faster than training and evaluating different models multiple times, as our experiments show in Section 5.4. Further, model prediction process is fully automatic since it does not require users to choose or fine-tune any values at test time. Figure 2 shows an overview of the model prediction process, and Algorithm 1 in the Appendix lists steps for offline meta-training and online model prediction.

## 4.3 STRUCTURAL META-GRAPH FEATURES

Meta-graph features are a crucial component of our meta-learning approach METAGL since they capture important structural characteristics of an arbitrary graph. Meta-graph features enable METAGL to quantify graph similarities, and utilize prior experience with observed graphs for GL model selection. It is important that a sufficient and representative set of meta-graph features are used to capture the important structural properties of graphs from a wide variety of domains, including biological, technological, information, and social networks to name a few.

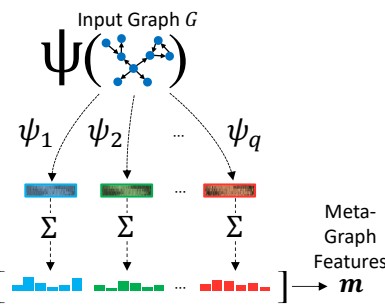

Figure 3: Meta-graph features in METAGL are derived in two steps. See Section 4.3 for details.

In this work, we are not able to leverage the commonly used simple statistical meta-features used by previous work on model selection-based meta-learning, as they cannot be used directly

over irregular and complex graph data. To address this problem, we introduce the notion of meta-graph features and develop a general framework for computing them on any arbitrary graph.

Meta-graph features in METAGL are derived in two steps, which is shown in Figure 3. First, we apply a set of structural meta-feature extractors $\Psi = \{\psi_1, \ldots, \psi_q\}$ to the input graph $G$, obtaining $\Psi(G) = \{\psi_1(G), \ldots, \psi_q(G)\}$. Applying $\psi \in \Psi$ to $G$ yields a vector or a distribution of values for the nodes (or edges) in the graph, such as degree distribution and PageRank scores. That is, in Figure 3, $\psi_1$ can be a degree distribution, $\psi_2$ can be PageRank scores of all nodes, and so on. Specifically, we use both local and global structural feature extractors. To capture the local structural properties around a node or an edge, we compute node degree, number of wedges (*i.e.*, a path of length 2), triangles centered at each node, and also frequency of triangles for every edge. To capture global structural properties of a node, we derive eccentricity, PageRank score, and k-core number of each node. Appendix D summarizes meta-feature extractors used in this work.

Let $\psi$ denote the local structural extractors for nodes. Given a graph $G_i = (V_i, E_i)$ and $\psi$, we obtain an $|V_i|$-dimensional node vector $\psi(G_i)$. Since any two graphs $G_i$ and $G_j$ are likely to have a different number of nodes and edges, the resulting structural feature matrices $\psi(G_i)$ and $\psi(G_j)$ for these graphs are also likely to be of different sizes as the rows of these matrices correspond to nodes or edges of the corresponding graph. Thus, in general, these structural feature-based representations of the graphs cannot be used directly to derive similarity between graphs.

Now, to address this issue, we apply the set $\Sigma$ of *global* statistical meta-graph feature extractors to every $\psi_i(G)$, $\forall i = 1, \ldots, q$, which summarizes each $\psi_i(G)$ to a fixed-size vector. Specifically, $\Sigma(\psi_i(G))$ applies each of the statistical functions in $\Sigma$ (*e.g.*, mean, kurtosis, etc) to the distribution $\psi_i(G)$, which computes a real number that summarizes the given feature distribution $\psi_i(G)$ from different statistical point of view, producing a vector $\Sigma(\psi_i(G)) \in \mathbb{R}^{|\Sigma|}$. Then we obtain the meta-graph feature vector $\mathbf{m}$ of graph $G$ by concatenating the resulting meta-graph feature vectors:

$$\mathbf{m} = [\Sigma(\psi_1(G)) \ \cdots \ \Sigma(\psi_q(G))] \in \mathbb{R}^d. \tag{6}$$

Table 5 in Appendix D lists the global statistical functions $\Sigma$ used in this work to derive meta-graph features. Further, in addition to the node- and edge-level structural features, we also compute global graph statistics (scalars directly derived from the graph, *e.g.*, density and degree assortativity coefficient), and append them to $\mathbf{m}$, *i.e.*, the node- or edge-level structural features obtained above.

Most importantly, given any arbitrary graph $G'$, the proposed approach is guaranteed to output a fixed $d$-dimensional meta-graph feature vector characterizing $G'$. Hence, the structural similarity of any two graphs $G$ and $G'$ can be quantified using a similarity function over $\mathbf{m}$ and $\mathbf{m}'$, respectively.

## 4.4 EMBEDDING MODELS AND GRAPHS

Given an informative context (*i.e.*, input features) of models and graphs that METAGL learns from model performances $\mathbf{P}$ and meta-graph features $\mathbf{M}$ (Sections 4.1 and 4.3), how can we use it to effectively learn model and graph embeddings that capture graph-to-model affinity? We note that similar entities can make each other's context more accurate and informative. For instance, in our problem setup, similar models tend to have similar performance distributions over graphs, and likewise similar graphs are likely to exhibit similar affinity to different models. With this consideration, we model the task as a graph representation learning problem, where we construct a graph called *G-M network* that connects similar graphs and models, and learn the graph and model embeddings over it.

**G-M Network.** We define G-M network to be a multi-relational graph with two types of nodes (*i.e.*, models and graphs) where edges connect similar model nodes and graph nodes. To measure similarity among graphs and models, we utilize the latent graph and model factors ($\mathbf{U}$ and $\mathbf{V}$, respectively) obtained by factorizing $\mathbf{P}$, as well as the meta-graph features $\mathbf{M}$. More precisely, we use the estimated graph factor $\widehat{\mathbf{U}}$ instead of $\mathbf{U}$ to let the same graph construction process work for new graphs. Note that this gives us two types of features for graph nodes (*i.e.*, $\widehat{\mathbf{U}}$ and $\mathbf{M}$), and one type of features for model nodes (*i.e.*, $\mathbf{V}$). To let different features influence the embedding step differently as needed, we connect graph nodes and model nodes using five type of edges: M-g2g, P-g2g, P-m2m, P-g2m, P-m2g where g and m denote the type of nodes that an edge connects (graph and model, respectively), and M and P denote that the edge is based on meta-graph features and model performance, respectively. For example, M-g2g and P-g2g edges connect two graph nodes that are similar in terms of $\mathbf{M}$ and $\widehat{\mathbf{U}}$, respectively. Then for each edge type, we construct a $k$-NN graph by connecting nodes to their top-$k$ similar nodes, where node-to-node similarity is defined as the cosine similarity between the

corresponding node features. For instance, for P-g2m edge type, graph nodes and model nodes are linked based on the similarity between $\widehat{\mathbf{U}}$ and $\mathbf{V}$. Fig. 7 in the Appendix illustrates the G-M network.

**Learning Over G-M Network.** Given the G-M network $\mathcal{G}_{\text{train}}$ with meta-train graphs and models, graph neural networks (GNNs) provide an effective framework to embed models and graphs via (weighted) neighborhood aggregation over $\mathcal{G}_{\text{train}}$. However, since the structure of G-M network is induced by simple $k$-NN search, some of the neighbors may not provide the same amount of information as others, or may even provide noisy information. We found it helpful to perform attentive neighborhood aggregation, so more informative neighbors can be given more weights. To this end, we choose to use attentive GNNs designed for multi-relational networks, and specifically use HGT (Hu et al., 2020). Then the embedding function $f(\cdot)$ in Section 4.1 is defined to be $f(\mathbf{x}) = \text{HGT}(\mathbf{x}, \mathcal{G}_{\text{train}})$ during training, which transforms the input node feature $\mathbf{x}$ into an embedding via attentive neighborhood aggregation over $\mathcal{G}_{\text{train}}$. Further details of HGT are provided in Appendix G.4.

**Inference Over G-M Network.** For inference at test time, we extend $\mathcal{G}_{\text{train}}$ to be a larger G-M network $\mathcal{G}_{\text{test}}$ that additionally contains test graph nodes, and edges between test graph nodes, and existing graphs and models in $\mathcal{G}_{\text{train}}$. The extension is done in the same way as in the training phase, by finding top-$k$ similar nodes. Then the embedding at test time can be done by $f(\mathbf{x}) = \text{HGT}(\mathbf{x}, \mathcal{G}_{\text{test}})$.

## 5 EXPERIMENTS

### 5.1 EXPERIMENTAL SETTINGS

**Models and Evaluation.** A model in our problem (Eqn. 1) consists of two components. The first component performs graph representation learning (GRL), and the other component leverages the learned embeddings for a downstream task of interest. In this work, we focus on link prediction, which is a key task for graph-structured data as we discuss in Section 1. We evaluate the performance of selecting a link prediction model for new graphs *without any model evaluation*. For the first component, we use 12 popular GRL methods, and for the second component for link scoring, we use a simple estimator that computes the cosine similarity between two node embeddings. This results in a model set $\mathcal{M}$ with 423 models. The full list of models is given in Table 4 in Appendix C.

For evaluation, we create a testbed containing benchmark graphs, meta-graph features, and a performance matrix. We construct the performance matrix by evaluating each link prediction model in $\mathcal{M}$ on the graphs in the testbed, in terms of mean average precision score. Then we evaluate METAGL and baselines via 5-fold cross validation where the benchmark graphs are split into meta-train $\mathcal{G}_{\text{train}}$ and meta-test $\mathcal{G}_{\text{test}}$ graphs for each fold, and meta-learners trained over the meta-train graph data are evaluated using the meta-test graph datasets. Thus, model performances over the meta-test graphs $\mathcal{G}_{\text{test}}$ and the meta-graph features of $\mathcal{G}_{\text{test}}$ were unseen during training, but used only for testing.

Since model selection aims to accurately predict the best model for a new graph, we evaluate the top-1 prediction performance in terms of MRR (Mean Reciprocal Rank), AUC, and NDCG (Normalized Discounted Cumulative Gain). To apply MRR and AUC, we label models such that the top-1 model (*i.e.*, the model with the best performance for the given graph) is labeled as 1, while all others are labeled as 0. For NDCG, we report NDCG@1, which evaluates the relevance of top-1 predicted model. All metrics range from 0 to 1, with larger values indicating better performance.

**Baselines.** Being the first work for evaluation-free model selection in GL, we do not have immediate baselines for comparison. Instead, we adapt baselines used for OD model selection (Zhao et al., 2021) and collaborative filtering for our problem setting. In Appendix A, we describe baselines in detail. Baselines are grouped into two categories: (a) *Simple meta-learners* select a model that performs

Table 2: The proposed METAGL outperforms existing meta-learners, given fully observed performance matrix. **Best** results are in bold, and second best results are underlined. "METAGL_Baseline" notation (*e.g.*, METAGL_S2) indicate that the baseline meta-learner uses METAGL's meta-graph features.

| | Method | MRR | AUC | NDCG@1 |
|---|---|---|---|---|
| | Random Selection | 0.011 | 0.490 | 0.745 |
| *Simple* | Global Best-AvgPerf | 0.163 | 0.877 | 0.932 |
| | Global Best-AvgRank | 0.103 | 0.867 | 0.930 |
| | METAGL_AS | 0.222 | 0.905 | 0.947 |
| | METAGL_ISAC | 0.202 | 0.887 | 0.939 |
| *Optimization-based* | METAGL_S2 | 0.170 | 0.910 | 0.945 |
| | METAGL_ALORS | 0.190 | 0.897 | 0.950 |
| | METAGL_NCF | 0.140 | 0.869 | 0.934 |
| | METAGL_MetaOD | 0.075 | 0.599 | 0.889 |
| | METAGL | **0.259** | **0.941** | **0.962** |

generally well, either globally or locally: **Global Best (GB)-AvgPerf**, **GB-AvgRank**, **ISAC** (Kadioglu et al., 2010), and **ARGOSMART (AS)** (Nikolić et al., 2013); (b) *Optimization-based meta-*

*learners* learn to estimate the model performance by modeling the relation between meta features and model performances: **Supervised Surrogates (S2)** (Xu et al., 2012), **ALORS** (Misir & Sebag, 2017), **NCF** (He et al., 2017), and **MetaOD** (Zhao et al., 2021). We also include **Random Selection (RS)** as a baseline to see how these methods compare to random scoring.

Note that, except the simplest meta-learners RS and GB, no baselines can handle graph data, and thus they cannot estimate model performance on the new graph on their own. In that sense, they are not direct competitors of METAGL. We enable them to be used for GL model selection by providing the proposed meta-graph features. MetaOD, which was originally designed for OD model selection, is also given the same meta-graph features to perform GL model selection. We denote baselines either by combining METAGL with their names (*e.g.*, METAGL_S2) to clearly show that they use METAGL's meta-graph features, or using their name alone (*e.g.*, S2) for simplicity.

## 5.2 MODEL SELECTION ACCURACY

**Fully Observed Performance Matrix.** In this setup, meta-learners are trained using a full performance matrix $\mathbf{P}$ with no missing entries. The model selection accuracy of all meta-learners in this setup is reported in Table 2, where METAGL achieves the best performance in all metrics, with *17%* higher MRR than the best baseline (AS).

- Among simple meta-learners, Global Best meta-learners, which simply average model performance or rank over all observed graphs, are outperformed by more sophisticated meta-learners AS and ISAC, which leverage dataset similarities for model selection using meta-graph features.
- For optimization-based meta-learners, it is important to be aware of how models and graphs relate to each other, and have high flexibility to capture that complex relationship. In methods like ALORS and MetaOD, relations between models and datasets (*i.e.*, relative position of models and datasets in the embedding subspace) are learned rather indirectly via reconstructing the performance matrix.
- METAGL, in contrast, directly captures graph-to-model affinity by modeling their relations via employing flexible GNNs over the G-M network, as well as reconstructing the performance matrix. As a result, METAGL consistently outperforms other optimization-based meta-learners.

**Partially Observed Performance Matrix.** In this setup, meta-learners are trained using a sparse performance matrix $\mathbf{P}$, obtained by randomly masking out a full $\mathbf{P}$. Figure 4 reports results obtained with varying sparsity, ranging up to 0.9. In this more challenging setup, METAGL consistently performs the best across all levels of sparsity, achieving up to *47%* higher MRR than the best baseline.

- With increased sparsity, nearly all meta-learners perform increasingly worse, as one might expect.
- While AS was the best baseline given a full $\mathbf{P}$, its accuracy decreased rapidly as sparsity increased. Since AS selects a model based on the 1NN meta-train graph, it is highly sensitive to $\mathbf{P}$'s sparsity.
- Baselines such as the Global Best baselines are more stable as they average across multiple graphs.
- Optimization-based methods like METAGL and S2 perform favorably to simple meta-learners in sparse settings as they learn to reconstruct $\mathbf{P}$ by modeling the relation between graphs and models.

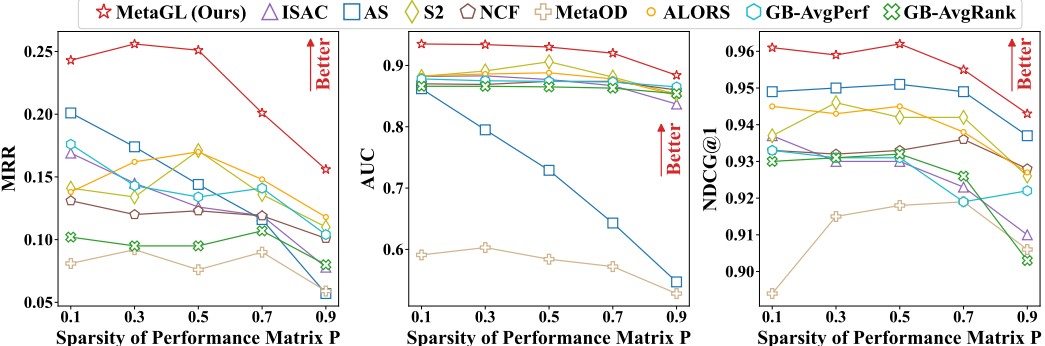

Figure 4: METAGL consistently performs the best, as the sparsity of performance matrix $\mathbf{P}$ varies. METAGL and all baseline meta-learners used the same meta-graph features proposed in this work.

## 5.3 EFFECTIVENESS OF META-GRAPH FEATURES

In Figure 5, we evaluate how the performance of meta-learners obtained with the proposed meta-graph feature (Section 4.3) compares to that obtained with existing graph-level embedding (GLE) techniques, GL2Vec (Chen & Koga, 2019), Graph2Vec (Narayanan et al., 2017), and GraphLoG (Xu et al., 2021).

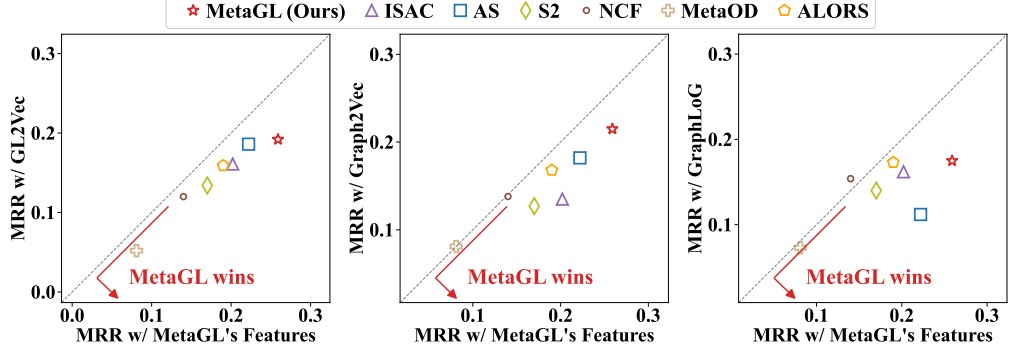

Figure 5: Using the proposed meta-graph features (Section 4.3), all meta-learners nearly consistently perform better (*i.e.*, points are below the diagonal) than with existing graph-level embedding methods.

Figure 8 in App. I.1 provides results for three other GLE methods, WaveletCharacteristic (Wang et al., 2021), SF (de Lara & Pineau, 2018), and LDP (Cai & Wang, 2019).

- Most points are below the diagonal in Figure 5, *i.e.*, all meta-learners nearly consistently perform better when they use METAGL's meta-graph features than when they use existing GLE methods. This shows the effectiveness of METAGL's features for the proposed task of GL model selection.
- METAGL performs the best, whether METAGL's features or existing GLE methods are used.

### 5.4 MODEL SELECTION EFFICIENCY

To evaluate how efficient METAGL's model selection is, we measure its runtime (*i.e.*, the time to create meta-graph features for the new graph at test time, plus the time to predict the best model), and compare it with the time to train a GL model. Figure 6 shows the distribution in box plots, where red and green lines denote the median and mean, respectively.

Results show that METAGL is fast, and incurs negligible runtime overhead: its runtime is just around 1 seconds or less in most cases (Figure 6a). Notably, compared to training each GL model for only 5% of its available model configurations, METAGL takes considerably less

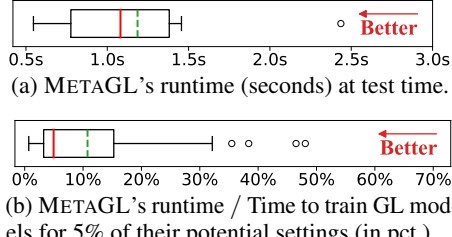

(a) METAGL's runtime (seconds) at test time.

(b) METAGL's runtime / Time to train GL models for 5% of their potential settings (in pct.).

Figure 6: METAGL is fast (∼1 sec.), and incurs negligible overhead. Red and green lines denote the median and mean, respectively.

time, *i.e.*, a median of 5% and a mean of 11% of the time required for model training (Figure 6b). Given large-scale test graphs in practice, the speed-up enabled by METAGL will be greater than that reported in Figure 6b, due to the increased training time on such graphs. Also, METAGL's model selection process can be further streamlined, *e.g.*, by parallelizing meta-feature generation process. We provide additional results on the runtime of METAGL and naive model selection in Appendix I.2.

### 5.5 ADDITIONAL RESULTS

We present ablation study in Appendix I.3, which shows the effectiveness of METAGL's proposed components, *e.g.*, the meta-learning objective, G-M network, and the graph encoder used by METAGL. We evaluate the sensitivity of model selection approaches to the variance of performance matrix **P** in Appendix I.4, and compare the predicted model performance with the actual best performance in Appendix I.5.

### 6 CONCLUSION

As more and more GL models are developed, selecting which one to use is becoming increasingly hard. Toward near-instantaneous, automatic GL model selection, we make the following contributions.

- **Problem Formulation.** We present the first problem formulation to select effective GL models in an evaluation-free manner (*i.e.*, without ever having to train/evaluate any model on the new graph).
- **Meta-Learning Framework and Features.** We propose METAGL, the first meta-learning framework for evaluation-free GL model selection, and meta-graph features to quantify graph similarities.
- **Effectiveness.** Using METAGL for model selection greatly outperforms existing meta-learning techniques (up to *47%* better), while incurring negligible runtime overhead at test time (∼*1 sec*).

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

## A  BASELINES

Being the first work for evaluation-free model selection in graph learning (GL), we do not have immediate baselines for comparison. Instead, we adapt baselines used in MetaOD (Zhao et al., 2021) for outlier detection (OD) model selection as well as collaborative filtering for our problem setting. The baselines used in experiments can be organized into the following two categories.

(a) ***Simple meta-learners*** select a model that performs generally well, either globally or locally.

- **Global Best (GB)-AvgPerf** selects the model that has the largest average performance over all meta-train graphs.
- **Global Best (GB)-AvgRank** computes the rank of all models (in percentile) for each graph, and selects the model with the largest average ranking over all meta-train graphs.
- **ISAC** (Kadioglu et al., 2010) first clusters meta-train datasets using meta-graph features, and at test time, finds the cluster closest to the test graph, and selects the model with the largest average performance over all graphs in that cluster.
- **ARGOSMART (AS)** (Nikolić et al., 2013) finds the meta-train graph closest to the test graph (*i.e.*, 1NN) in terms of meta-graph feature similarity, and selects the model with the best result on the 1NN graph.

(b) ***Optimization-based meta-learners*** learn to estimate the model performance by modeling the relation between meta-graph features and model performances.

- **Supervised Surrogates (S2)** (Xu et al., 2012) learns a surrogate model (a regressor) that maps meta-graph features to model performances.
- **ALORS** (Misir & Sebag, 2017) factorizes the performance matrix into latent factors on graphs and models, and estimates the performance to be the dot product between the two factors, where a non-linear regressor maps meta-graph features into the latent graph factors.
- **NCF** (He et al., 2017) replaces the dot product used in ALORS with a more general neural architecture that estimates performance by combining the linearity of matrix factorization and non-linearity of deep neural networks.
- **MetaOD** (Zhao et al., 2021) pioneered the field of unsupervised OD model selection by designing meta-features specialized to capture the outlying characteristics of datasets, as well as improving upon ALORS with the adoption of an NDCG-based meta-training objective. We enable MetaOD to be applicable to our problem setting, by applying our proposed meta-graph features to MetaOD.

In addition, we also include **Random Selection (RS)** as a baseline, to see how meta-learners perform in comparison to random scoring. Note that among the above approaches, only GB-AvgPerf and GB-AvgRank do not rely on meta-features for model selection. All other meta-learners make use of the proposed meta-graph features to estimate model performances on an unseen test graph.

## B  NOTATIONS

Table 3 provides a list of notations frequently used in this work.

## C  MODEL SET

A model in the model set $\mathcal{M}$ refers to a graph representation learning (GRL) method along with its hyperparameters settings, and a predictor that makes a downstream task-specific prediction given the node embeddings from the GRL method. In this work, we use a link predictor which scores a given link by computing the cosine similarity between the two nodes' embeddings. Table 4 shows the

Table 3: Summary of notations.

| | |
|---|---|
| $\mathcal{G}$ | set of graphs $\{G_1, \ldots, G_n\}$ in the training set |
| $n$ | number of graphs in training set $n = |\mathcal{G}|$ |
| $G_{\text{test}}$ | new unseen test graph $G_{\text{test}} \notin \mathcal{G}$ |
| $\mathcal{M}$ | model set $\{M_1, \ldots, M_m\}$ to search over |
| $m$ | number of models to search over $m = |\mathcal{M}|$ |
| $\Psi$ | set of structural meta-node/edge feature extractors |
| $\Sigma$ | set of meta-graph feature extractors |
| $\mathbf{M}$ | meta-graph feature matrix where $\mathbf{M} \in \mathbb{R}^{n \times d}$ |
| $d$ | number of meta-graph features |
| $\mathbf{m}_{\text{test}}$ | meta-graph feature vector for the new unseen test graph $G_{\text{test}}$ |
| $k$ | embedding size |
| $\mathbf{P}$ | performance matrix of $m$ models on $n$ graphs |
| $\mathbf{U}$ | latent graph factors obtained by factorizing $\mathbf{P}$ ($\mathbf{P} \approx \mathbf{U}\mathbf{V}^\intercal$) |
| $\mathbf{V}$ | latent model factors obtained by factorizing $\mathbf{P}$ ($\mathbf{P} \approx \mathbf{U}\mathbf{V}^\intercal$) |
| $f(\cdot)$ | learnable embedding function for models and graphs |

Table 4: Graph representation learning (GRL) models and their hyperparameter settings, which collectively comprise the model set $\mathcal{M}$ with 423 unique GRL models. For details of the hyperparameters, please refer to the cited paper.

| Methods | Hyperparameter Settings | Count |
|---|---|---|
| SGC (Wu et al., 2019a) | # (number of) hops $k \in \{1, 2, 3\}$ | 3 |
| GCN (Kipf & Welling, 2017) | # layers $L \in \{1, 2, 3\}$, # epochs $N \in \{1, 10\}$ | 6 |
| GraphSAGE (Hamilton et al., 2017) | # layers $L \in \{1, 2, 3\}$, # epochs $N \in \{1, 10\}$, aggregation functions $f \in \{\text{mean, gcn, lstm}\}$ | 18 |
| node2vec (Grover & Leskovec, 2016) | $p, q \in \{1, 2, 4\}$ | 9 |
| role2vec (Ahmed et al., 2018) | $p, q \in \{0.25, 1, 4\}$, $\alpha \in \{0.01, 0.1, 0.5, 0.9, 0.99\}$, motif combinations $\mathcal{H} \in \{\{H_1\}, \{H_2, H_3\}, \{H_2, H_3, H_4, H_6, H_8\}, \{H_1, H_2, \ldots, H_8\}\}$ | 180 |
| GraRep (Cao et al., 2015) | $k \in \{1, 2\}$ | 2 |
| DeepWalk (Perozzi et al., 2014) | $p = 1, q = 1$ | 1 |
| HONE (Rossi et al., 2018) | $k \in \{1, 2\}$, $D_{\text{local}} \in \{4, 8, 16\}$, variant $v = \{1, 2, 3, 4, 5\}$ | 30 |
| node2bits (Jin et al., 2019) | walk num $w_n \in \{5, 10, 20\}$, walk len $w_l \in \{5, 10, 20\}$, log base $b \in \{2, 4, 8, 10\}$, feats $f \in \{16\}$ | 36 |
| DeepGL (Rossi et al., 2020) | $\alpha \in \{0.1, 0.3, 0.5, 0.7, 0.9\}$, motif size $\in \{4\}$, eps tolerance $t \in \{0.01, 0.05, 0.1\}$, relational aggr. $\in \{\{m\}, \{p\}, \{s\}, \{v\}, \{m, p\}, \{m, v\}, \{s, m\}, \{s, p\}, \{s, v\}\}$ where $m, p, s, v$ denote mean, product, sum, var | 135 |
| LINE (Tang et al., 2015) | # hops/order $k \in \{1, 2\}$ | 2 |
| Spectral Emb. (Luo et al., 2003) | tolerance $t \in \{0.001\}$ | 1 |
| ***Total Count*** | | **423** |

complete list of 12 popular GRL methods and their specific hyperparameter settings, which compose 412 unique models in the model set $\mathcal{M}$. Note that the link predictor is omitted from Table 4 since we employ the same link predictor based on cosine similarity to all GRL methods.

# D   META-GRAPH FEATURES

**Structural Meta-Feature Extractors.** To capture the local structural properties around a node or an edge, we compute the distribution of node degrees, number of wedges (*i.e.*, a path of length 2), triangles centered at each node, as well as the frequency of triangles for each edge. To capture the global structural properties of a node, we derive the eccentricity, PageRank score, and k-core number of each node. We also capture the global graph-level statistics (*i.e.*, different from local

node/edge-level structural properties), such as the density of $\mathbf{A}$ and $\mathbf{A}\mathbf{A}^T$ where $\mathbf{A}$ is the adjacency matrix, and also the degree assortativity coefficient $r$.

**Global Statistical Functions.** For each of the structural property distributions (degree, k-core numbers, and so on) derived by the above structural meta-feature extractors, we apply the set $\Sigma$ of global statistical functions (Table 5) over it to obtain a fixed-length vector representation for the node/edge/graph-level structural feature distribution.

After obtaining a set of meta-graph features, we concatenate all of them together to create the final meta-graph feature vector $\mathbf{m}$ for the graph.

Table 5: Summary of the global statistical functions $\Sigma$ for deriving a set of meta-graph features from a graph invariant (*e.g.*, k-core numbers, node degrees, and so on). Let $\mathbf{x}$ denote an arbitrary graph invariant vector for some graph $G_i = (V_i, E_i)$ and $\pi(\mathbf{x})$ is the sorted vector of $\mathbf{x}$. Note $\mathbf{x}$ can be any representation, *e.g.*, node degree vector (value for each node in $G_i$) or a degree distribution vector.

| Name | Equation |
|------|----------|
| Num. unique values | $\mathrm{card}(\mathbf{x})$ |
| Density | $\mathrm{nnz}(\mathbf{x})/|\mathbf{x}|$ |
| $Q_1, Q_3$ | median of the $|\mathbf{x}|/2$ smallest (largest) values |
| IQR | $Q_3 - Q_1$ |
| Outlier LB $\alpha \in \{1.5, 3\}$ | $\sum_i \mathbb{I}(x_i < Q_1 - \alpha IQR)$ |
| Outlier UB $\alpha \in \{1.5, 3\}$ | $\sum_i \mathbb{I}(x_i > Q_3 + \alpha IQR)$ |
| Total outliers $\alpha \in \{1.5, 3\}$ | $\sum_i \mathbb{I}(x_i < Q_1 - \alpha IQR) + \sum_i \mathbb{I}(x_i > Q_3 + \alpha IQR)$ |
| ($\alpha$-std) outliers $\alpha \in \{2, 3\}$ | $\mu_\mathbf{x} \pm \alpha\sigma_\mathbf{x}$ |
| Spearman ($\rho$, p-val) | $\mathrm{spearman}(\mathbf{x}, \pi(\mathbf{x}))$ |
| Kendall ($\tau$, p-val) | $\mathrm{kendall}(\mathbf{x}, \pi(\mathbf{x}))$ |
| Pearson ($r$, p-val) | $\mathrm{pearson}(\mathbf{x}, \pi(\mathbf{x}))$ |
| Min, max | $\min(\mathbf{x}), \max(\mathbf{x})$ |
| Range | $\max(\mathbf{x}) - \min(\mathbf{x})$ |
| Median | $\mathrm{med}(\mathbf{x})$ |
| Geometric Mean | $|\mathbf{x}|^{-1} \prod_i x_i$ |
| Harmonic Mean | $|\mathbf{x}| / \sum_i \frac{1}{x_i}$ |
| Mean, Stdev, Variance | $\mu_\mathbf{x}, \sigma_\mathbf{x}, \sigma_\mathbf{x}^2$ |
| Skewness | $\mathbb{E}(\mathbf{x}-\mu_\mathbf{x})^3/\sigma_\mathbf{x}^3$ |
| Kurtosis | $\mathbb{E}(\mathbf{x}-\mu_\mathbf{x})^4/\sigma_\mathbf{x}^4$ |
| Quartile Dispersion Coeff. | $\frac{Q_3-Q_1}{Q_3+Q_1}$ |
| Median Absolute Deviation | $\mathrm{med}(|\mathbf{x} - \mathrm{med}(\mathbf{x})|)$ |
| Avg. Absolute Deviation | $\frac{1}{|\mathbf{x}|}\mathbf{e}^T|\mathbf{x} - \mu_\mathbf{x}|$ |
| Coeff. of Variation | $\sigma_\mathbf{x}/\mu_\mathbf{x}$ |
| Efficiency ratio | $\sigma_\mathbf{x}^2/\mu_\mathbf{x}^2$ |
| Variance-to-mean ratio | $\sigma_\mathbf{x}^2/\mu_\mathbf{x}$ |
| Signal-to-noise ratio (SNR) | $\mu_\mathbf{x}^2/\sigma_\mathbf{x}^2$ |
| Entropy | $H(\mathbf{x}) = -\sum_i x_i \log x_i$ |
| Norm. entropy | $H(\mathbf{x})/\log_2|\mathbf{x}|$ |
| Gini coefficient | $-$ |
| Quartile max gap | $\max(Q_{i+1} - Q_i)$ |
| Centroid max gap | $\max_{ij} |c_i - c_j|$ |
| Histogram prob. dist. | $\mathbf{p}_h = \frac{\mathbf{h}}{\mathbf{h}^T\mathbf{e}}$ (with fixed # of bins) |

# E  GRAPH DATASETS

The testbed used in this work comprises 301 graphs that have widely varying structural characteristics. Table 6 provides a list of all graph datasets in the testbed. All graph data are from (Rossi & Ahmed, 2015); they are publicly available under the Creative Commons Attribution-ShareAlike License.

Table 6: Summary of the 301 graph datasets that comprise the testbed.

| | Graph | # Nodes | # Edges | | | Graph | # Nodes | # Edges |
|---|---|---|---|---|---|---|---|---|
| 1 | BA-1_10_60 | 804 | 46410 | 152 | | enzymes_g297 | 121 | 149 |
| 2 | BA-1_11_40 | 917 | 35860 | 153 | | enzymes_g349 | 64 | 118 |
| 3 | BA-1_12_10 | 827 | 8215 | 154 | | enzymes_g355 | 66 | 112 |
| 4 | BA-1_13_10 | 1043 | 10375 | 155 | | enzymes_g504 | 66 | 120 |
| 5 | BA-1_14_40 | 829 | 32340 | 156 | | enzymes_g523 | 48 | 111 |
| 6 | BA-1_16_40 | 1126 | 44220 | 157 | | enzymes_g526 | 58 | 110 |
| 7 | BA-1_17_10 | 895 | 8895 | 158 | | enzymes_g527 | 57 | 107 |
| 8 | BA-1_18_40 | 1141 | 44820 | 159 | | enzymes_g532 | 74 | 120 |
| 9 | BA-1_1_10 | 862 | 8565 | 160 | | enzymes_g575 | 51 | 110 |
| 10 | BA-1_20_10 | 830 | 8245 | 161 | | enzymes_g578 | 60 | 103 |
| 11 | BA-1_22_10 | 1094 | 10885 | 162 | | enzymes_g594 | 52 | 114 |
| 12 | BA-1_24_40 | 946 | 37020 | 163 | | enzymes_g597 | 52 | 116 |
| 13 | BA-1_3_40 | 1017 | 39860 | 164 | | enzymes_g598 | 55 | 100 |
| 14 | BA-1_4_40 | 970 | 37980 | 165 | | enzymes_g8 | 88 | 133 |
| 15 | BA-1_5_10 | 1050 | 10445 | 166 | | ER-1_11_5 | 917 | 20841 |
| 16 | BA-1_6_60 | 803 | 46350 | 167 | | ER-1_14_5 | 829 | 17092 |
| 17 | BA-1_7_40 | 832 | 32460 | 168 | | ER-1_15_5 | 826 | 17099 |
| 18 | BA-1_8_10 | 1040 | 10345 | 169 | | ER-1_24_5 | 946 | 22383 |
| 19 | bio-CE-GN | 2220 | 53683 | 170 | | ER-1_4_5 | 970 | 23747 |
| 20 | bio-CE-GT | 924 | 3239 | 171 | | ER-2_6_5 | 8115 | 16868 |
| 21 | bio-CE-HT | 2617 | 2985 | 172 | | inf-euroroad | 1174 | 1417 |
| 22 | bio-CE-LC | 1387 | 1648 | 173 | | inf-openflights | 2939 | 15677 |
| 23 | bio-CE-PG | 1871 | 47754 | 174 | | KPGM-log10-10-trial1 | 845 | 7194 |
| 24 | bio-DM-CX | 4040 | 76717 | 175 | | KPGM-log10-10-trial2 | 830 | 7265 |
| 25 | bio-DM-HT | 2989 | 4660 | 176 | | KPGM-log10-10-trial3 | 837 | 7251 |
| 26 | bio-DM-LC | 658 | 1129 | 177 | | KPGM-log10-12-trial1 | 845 | 8467 |
| 27 | bio-DR-CX | 3289 | 84940 | 178 | | KPGM-log10-12-trial2 | 864 | 8339 |
| 28 | bio-grid-fission-yeast | 2026 | 12637 | 179 | | KPGM-log10-12-trial3 | 856 | 8393 |
| 29 | bio-HS-CX | 4413 | 108818 | 180 | | KPGM-log10-14-trial2 | 885 | 9405 |
| 30 | bio-HS-HT | 2570 | 13691 | 181 | | KPGM-log10-8-trial1 | 824 | 6055 |
| 31 | bio-HS-LC | 4227 | 39484 | 182 | | KPGM-log10-8-trial2 | 796 | 6040 |
| 32 | bio-SC-CC | 2223 | 34879 | 183 | | KPGM-log10-8-trial3 | 813 | 6052 |
| 33 | bio-SC-GT | 1716 | 33987 | 184 | | KPGM-log8-10-trial1 | 220 | 1502 |
| 34 | bio-SC-HT | 2084 | 63027 | 185 | | KPGM-log8-10-trial2 | 220 | 1518 |
| 35 | bio-SC-LC | 2004 | 20452 | 186 | | KPGM-log8-10-trial3 | 224 | 1520 |
| 36 | bio-SC-TS | 636 | 3959 | 187 | | KPGM-log8-12-trial1 | 224 | 1729 |
| 37 | biogrid-human | 2005 | 3959 | 188 | | KPGM-log8-12-trial2 | 229 | 1756 |
| 38 | biogrid-mouse | 1450 | 1636 | 189 | | KPGM-log8-12-trial3 | 230 | 1761 |
| 39 | biogrid-plant | 523 | 838 | 190 | | KPGM-log8-14-trial1 | 229 | 1947 |
| 40 | biogrid-worm | 1930 | 3576 | 191 | | KPGM-log8-14-trial2 | 233 | 1950 |
| 41 | biogrid-yeast | 836 | 1049 | 192 | | KPGM-log8-14-trial3 | 230 | 1943 |
| 42 | bn-cat-mixed-species-brain1 | 65 | 730 | 193 | | KPGM-log8-16-trial1 | 232 | 2117 |
| 43 | bn-fly-drosophila-medulla1 | 1781 | 9016 | 194 | | KPGM-log8-16-trial2 | 230 | 2152 |
| 44 | bn-macaque-rhesus-brain1 | 242 | 3054 | 195 | | KPGM-log8-16-trial3 | 235 | 2178 |
| 45 | bn-macaque-rhesus-brain2 | 91 | 582 | 196 | | KPGM-log8-8-trial1 | 223 | 1324 |
| 46 | bn-macaque-rhesus-cerebral-cortex1 | 91 | 1401 | 197 | | KPGM-log8-8-trial2 | 218 | 1312 |
| 47 | bn-macaque-rhesus-interareal-cortical2 | 93 | 2262 | 198 | | KPGM-log8-8-trial3 | 215 | 1303 |
| 48 | bn-mouse-brain1 | 213 | 16242 | 199 | | nci1_g1677 | 102 | 106 |
| 49 | bn-mouse-kasthuri-v4 | 1029 | 1559 | 200 | | nci1_g1863 | 107 | 111 |
| 50 | bn-mouse-visual-cortex2 | 193 | 214 | 201 | | nci1_g1893 | 96 | 102 |
| 51 | ca-AstroPh | 17903 | 196972 | 202 | | nci1_g1894 | 104 | 108 |
| 52 | ca-CondMat | 21363 | 91286 | 203 | | nci1_g2079 | 88 | 103 |
| 53 | ca-cora | 2708 | 5278 | 204 | | nci1_g2082 | 86 | 101 |
| 54 | ca-CSphd | 1882 | 1740 | 205 | | nci1_g2172 | 106 | 107 |
| 55 | ca-DBLP-kang | 2879 | 11326 | 206 | | nci1_g2228 | 92 | 98 |
| 56 | ca-Erdos992 | 5094 | 7515 | 207 | | nci1_g2229 | 92 | 98 |
| 57 | ca-GrQc | 4158 | 13422 | 208 | | nci1_g2443 | 91 | 97 |
| 58 | ca-HepPh | 11204 | 117619 | 209 | | nci1_g2455 | 90 | 96 |
| 59 | ca-netscience | 379 | 914 | 210 | | nci1_g3139 | 107 | 112 |
| 60 | ca-sandi-auths | 86 | 123 | 211 | | nci1_g3141 | 92 | 98 |
| 61 | CL-1000-1d7-trial1 | 928 | 4653 | 212 | | nci1_g3145 | 91 | 97 |
| 62 | CL-1000-1d7-trial2 | 932 | 4888 | 213 | | nci1_g3444 | 93 | 102 |
| 63 | CL-1000-1d7-trial3 | 938 | 4840 | 214 | | nci1_g3449 | 95 | 99 |
| 64 | CL-1000-1d8-trial1 | 925 | 3776 | 215 | | nci1_g3585 | 105 | 107 |
| 65 | CL-1000-1d8-trial2 | 930 | 4136 | 216 | | nci1_g3700 | 111 | 119 |

Table 6 – *Continued from the previous page*

| | Graph | # Nodes | # Edges | | | Graph | # Nodes | # Edges |
|---|---|---|---|---|---|---|---|---|
| 66 | CL-1000-1d8-trial3 | 925 | 3714 | | 217 | nci1_g3711 | 89 | 106 |
| 67 | CL-1000-1d9-trial1 | 928 | 3510 | | 218 | nci1_g3901 | 93 | 102 |
| 68 | CL-1000-1d9-trial2 | 912 | 3053 | | 219 | nci1_g3990 | 90 | 105 |
| 69 | CL-1000-1d9-trial3 | 932 | 3278 | | 220 | nci1_g4094 | 90 | 98 |
| 70 | CL-1000-2d0-trial1 | 909 | 2795 | | 221 | power-1138-bus | 1138 | 2596 |
| 71 | CL-1000-2d0-trial2 | 899 | 2941 | | 222 | power-494-bus | 494 | 1080 |
| 72 | CL-1000-2d0-trial3 | 916 | 3010 | | 223 | power-662-bus | 662 | 1568 |
| 73 | CL-1000-2d1-trial1 | 903 | 2430 | | 224 | power-685-bus | 685 | 1967 |
| 74 | CL-1000-2d1-trial2 | 911 | 2734 | | 225 | power-bcspwr09 | 1723 | 4117 |
| 75 | CL-1000-2d1-trial3 | 915 | 2782 | | 226 | power-eris1176 | 1176 | 9864 |
| 76 | DD_g1 | 327 | 899 | | 227 | rec-amazon | 91813 | 125704 |
| 77 | DD_g10 | 146 | 328 | | 228 | rec-movielens-tag-movies-10m | 16528 | 71081 |
| 78 | DD_g100 | 349 | 1005 | | 229 | road-chesapeake | 39 | 170 |
| 79 | DD_g1000 | 183 | 408 | | 230 | road-ChicagoRegional | 1467 | 1298 |
| 80 | DD_g1001 | 88 | 203 | | 231 | road-euroroad | 1174 | 1417 |
| 81 | DD_g1002 | 104 | 255 | | 232 | road-luxembourg-osm | 114599 | 119666 |
| 82 | DD_g1003 | 53 | 116 | | 233 | road-minnesota | 2642 | 3303 |
| 83 | DD_g1004 | 94 | 230 | | 234 | road-usroads-48 | 126146 | 161950 |
| 84 | DD_g1005 | 370 | 903 | | 235 | rt-retweet | 96 | 117 |
| 85 | DD_g1006 | 246 | 568 | | 236 | rt-twitter-copen | 761 | 1029 |
| 86 | DD_g1007 | 309 | 732 | | 237 | rt_alwefaq | 4171 | 7063 |
| 87 | DD_g1008 | 109 | 304 | | 238 | rt_assad | 2139 | 2788 |
| 88 | DD_g1009 | 129 | 272 | | 239 | rt_bahrain | 4676 | 7979 |
| 89 | DD_g101 | 306 | 728 | | 240 | rt_barackobama | 9631 | 9775 |
| 90 | DD_g1010 | 157 | 363 | | 241 | rt_damascus | 3052 | 3869 |
| 91 | DD_g1011 | 47 | 136 | | 242 | rt_dash | 6288 | 7436 |
| 92 | DD_g1012 | 146 | 365 | | 243 | rt_gmanews | 8373 | 8721 |
| 93 | DD_g1013 | 93 | 211 | | 244 | rt_gop | 4687 | 5529 |
| 94 | DD_g1014 | 119 | 273 | | 245 | rt_http | 8917 | 10314 |
| 95 | DD_g1015 | 102 | 244 | | 246 | rt_islam | 4497 | 4616 |
| 96 | DD_g1016 | 113 | 291 | | 247 | rt_israel | 3698 | 4165 |
| 97 | DD_g1017 | 162 | 376 | | 248 | rt_lebanon | 3961 | 4436 |
| 98 | DD_g1018 | 296 | 680 | | 249 | rt_libya | 5067 | 5541 |
| 99 | DD_g1019 | 131 | 353 | | 250 | rt_lolgop | 9765 | 10075 |
| 100 | DD_g102 | 561 | 1422 | | 251 | rt_obama | 3212 | 3423 |
| 101 | DD_g1020 | 228 | 541 | | 252 | rt_occupy | 3225 | 3944 |
| 102 | DD_g1021 | 329 | 787 | | 253 | rt_occupywallstnyc | 3609 | 3833 |
| 103 | DD_g1022 | 294 | 730 | | 254 | rt_oman | 4904 | 6230 |
| 104 | DD_g1023 | 172 | 425 | | 255 | rt_onedirection | 7987 | 8103 |
| 105 | DD_g1024 | 59 | 160 | | 256 | rt_p2 | 4902 | 6018 |
| 106 | DD_g1025 | 88 | 205 | | 257 | rt_saudi | 7252 | 8061 |
| 107 | DD_g1026 | 247 | 578 | | 258 | rt_tcot | 4547 | 5503 |
| 108 | DD_g1027 | 108 | 223 | | 259 | rt_tlot | 3665 | 4475 |
| 109 | DD_g1028 | 72 | 137 | | 260 | rt_uae | 5248 | 6387 |
| 110 | DD_g1029 | 99 | 215 | | 261 | rt_voteonedirection | 2280 | 2464 |
| 111 | DD_g103 | 265 | 647 | | 262 | sc-nasasrb | 54870 | 1311227 |
| 112 | DD_g1030 | 136 | 351 | | 263 | soc-advogato | 6551 | 43427 |
| 113 | DD_g104 | 372 | 999 | | 264 | soc-dolphins | 62 | 159 |
| 114 | DD_g105 | 423 | 1192 | | 265 | soc-firm-hi-tech | 33 | 91 |
| 115 | DD_g106 | 574 | 1355 | | 266 | soc-gplus | 23628 | 39194 |
| 116 | DD_g107 | 130 | 292 | | 267 | soc-hamsterster | 2426 | 16630 |
| 117 | DD_g108 | 483 | 1137 | | 268 | soc-highschool-moreno | 70 | 274 |
| 118 | DD_g109 | 132 | 315 | | 269 | soc-physicians | 241 | 923 |
| 119 | DD_g11 | 312 | 761 | | 270 | soc-sign-bitcoinalpha | 3783 | 14124 |
| 120 | DD_g110 | 394 | 1137 | | 271 | soc-student-coop | 185 | 311 |
| 121 | DD_g111 | 483 | 1520 | | 272 | soc-wiki-Vote | 889 | 2914 |
| 122 | DD_g112 | 266 | 631 | | 273 | socfb-Amherst | 2235 | 90954 |
| 123 | DD_g113 | 347 | 853 | | 274 | socfb-Bowdoin47 | 2252 | 84387 |
| 124 | DD_g114 | 334 | 761 | | 275 | socfb-Caltech | 769 | 16656 |
| 125 | DD_g115 | 336 | 946 | | 276 | socfb-Hamilton46 | 2314 | 96394 |
| 126 | eco-everglades | 69 | 885 | | 277 | socfb-Haverford76 | 1446 | 59589 |
| 127 | eco-florida | 128 | 2075 | | 278 | socfb-nips-ego | 2888 | 2981 |
| 128 | eco-foodweb-baydry | 128 | 2106 | | 279 | socfb-Oberlin44 | 2920 | 89912 |
| 129 | eco-foodweb-baywet | 128 | 2075 | | 280 | socfb-Reed98 | 962 | 18812 |
| 130 | eco-mangwet | 97 | 1446 | | 281 | socfb-Simmons81 | 1518 | 32988 |
| 131 | eco-stmarks | 54 | 353 | | 282 | socfb-Smith60 | 2970 | 97133 |
| 132 | email-dnc-corecipient | 906 | 10429 | | 283 | socfb-Swarthmore42 | 1659 | 61050 |
| 133 | email-dnc-leak | 1891 | 4465 | | 284 | socfb-Trinity100 | 2613 | 111996 |
| 134 | email-enron-only | 143 | 623 | | 285 | socfb-USFCA72 | 2682 | 65252 |
| 135 | email-EU | 32430 | 54397 | | 286 | socfb-Vassar85 | 3068 | 119161 |
| 136 | email-radoslaw | 167 | 3251 | | 287 | socfb-Villanova62 | 7772 | 314989 |
| 137 | email-univ | 1133 | 5451 | | 288 | socfb-Wellesley22 | 2970 | 94899 |
| 138 | enzymes_g103 | 59 | 115 | | 289 | socfb-Williams40 | 2790 | 112986 |
| 139 | enzymes_g118 | 95 | 121 | | 290 | tech-routers-rf | 2113 | 6632 |

*Continued on the next page*

Table 6 – *Continued from the previous page*

| | Graph | # Nodes | # Edges | | | Graph | # Nodes | # Edges |
|---|---|---|---|---|---|---|---|---|
| 140 | enzymes_g123 | 90 | 127 | | 291 | tech-routers-rf | 2113 | 6632 |
| 141 | enzymes_g199 | 62 | 108 | | 292 | web-BerkStan | 12305 | 19500 |
| 142 | enzymes_g204 | 57 | 105 | | 293 | web-edu | 3031 | 6474 |
| 143 | enzymes_g209 | 57 | 101 | | 294 | web-EPA | 4271 | 8909 |
| 144 | enzymes_g215 | 48 | 104 | | 295 | web-google | 1299 | 2773 |
| 145 | enzymes_g224 | 54 | 105 | | 296 | web-indochina-2004 | 11358 | 47606 |
| 146 | enzymes_g279 | 60 | 107 | | 297 | web-polblogs | 643 | 2280 |
| 147 | enzymes_g291 | 62 | 104 | | 298 | web-spam | 4767 | 37375 |
| 148 | enzymes_g292 | 60 | 100 | | 299 | web-webbase-2001 | 16062 | 25593 |
| 149 | enzymes_g293 | 96 | 109 | | 300 | web-wiki-chameleon | 2277 | 31421 |
| 150 | enzymes_g295 | 123 | 139 | | 301 | web-wiki-crocodile | 11631 | 170918 |
| 151 | enzymes_g296 | 125 | 141 | | | | | |

# F  EXPERIMENTAL DETAILS

## F.1  EXPERIMENTAL SETTINGS

**Software.** We used PyTorch[1] for implementing the training and inference pipeline, and used the DGL's implementation of HGT[2]. For MetaOD (Zhao et al., 2021), we used the implementation provided by the authors[3]. We used the Karate Club library (Rozemberczki et al., 2020) for the implementations of the following graph-level embedding (GLE) methods, Graph2Vec (Narayanan et al., 2017), GL2Vec (Chen & Koga, 2019), WaveletCharacteristic (Wang et al., 2021), SF (de Lara & Pineau, 2018), and LDP (Cai & Wang, 2019). For GraphLoG (Xu et al., 2021), we used the authors' implementation[4]. We used open source libraries, such as NetworkX[5] and NumPy[6], for implementing meta-graph feature extractors.

**Hyperparameters.** We set the embedding size $k$ to 32 for METAGL and other meta-learners that learn embeddings of models and graphs. For METAGL, we created the G-M network by connecting nodes to their top-30 similar nodes. As an the embedding function $f(\cdot)$ in METAGL, we used HGT (Hu et al., 2020) with 2 layers and 4 heads per layer. HGT is included in the Deep Graph Library (DGL), which is licensed under the Apache License 2.0. For training, we used the Adam optimizer with a learning rate of 0.00075 and a weight decay of 0.0001. For GLE approaches, we used the default hyperparameter settings specified in the corresponding library and GitHub repository.

**Link Prediction Model Training.** Given a graph $G$, we first hold out 10% of the edges in graph $G$ to be used for evaluation, and train GL models with the resulting subgraph for link prediction. The training of GL models was performed by sampling 20 negative edges per positive edge, computing the link score by applying a dot product between the two corresponding node embeddings, followed by a sigmoid function, and then optimizing a binary cross entropy loss for the positive and negative edge scores. For evaluation, we randomly sampled the same number of negative edges as the positive edges, and evaluated the predicted link scores in terms of mean average precision.

## F.2  EVALUATION OF MODEL SELECTION ACCURACY (SECTION 5.2)

In our evaluation involving a partially observed performance matrix, we extended baseline meta-learners as follows so they can operate in the presence of missing entries in the performance matrix.

- Global Best-AvgPerf averaged observed performance entries alone, ignoring missing values. If an average performance cannot be computed for some model (which is the case when a model has no observed performance entries for all graphs), we use the mean of averaged performance for other models in its place.

---

[1] https://pytorch.org/
[2] https://www.dgl.ai/
[3] https://github.com/yzhao062/MetaOD
[4] https://github.com/DeepGraphLearning/GraphLoG
[5] https://networkx.org/
[6] https://numpy.org/

- Global Best-AvgRank computed the model rankings for each graph in percentile, as the number of observed model performances may be different for different graphs, and averaged the rank percentiles for observed cases only, as in Global Best-AvgPerf.
- ISAC handled the sparse performance matrix in the same way as Global Best-AvgPerf, except that only a subset of graphs, which is similar to the test graph, is considered in ISAC.
- ARGOSMART (AS) computed the mean of observed performance entries of the 1NN graph, and used this quantity in place of missing values.
- ALORS factorized the sparse performance matrix using a missing value-aware non-negative matrix factorization algorithm.
- Supervised Sur. (S2), NCF, and MetaOD performed optimization by only considering observed performance values in the loss function, while skipping over missing entries. Early stopping based on the validation performance was also done with respect to the observed performances alone.

### F.3 EVALUATION OF META-GRAPH FEATURES (SECTION 5.3)

Except for WaveletCharacteristic and GraphLoG, we applied graph-level embedding (GLE) approaches to all graphs in our testbed, and meta-learners were trained and evaluated using the representations of all graphs via 5-fold cross validation. Since WaveletCharacteristic and GraphLoG could not scale up to some of the largest graphs in the testbed (*e.g.*, due to out-of-memory error), we excluded 9% and 27% largest graphs for GraphLoG and WaveletCharacteristic, respectively, and evaluated meta-learners using the resulting subset of graphs. Note that, in these cases, METAGL was also trained and evaluated using the same subset of graphs.

## G  ADDITIONAL DETAILS AND ANALYSIS OF METAGL

### G.1  METAGL ALGORITHM AND META-GRAPH FEATURES

Algorithm 1 provides detailed steps of METAGL, for both offline meta-training (top) and online model selection (bottom). In METAGL, we log-transform the meta-graph features, and extend the meta-graph features with them, as it helps with model selection. We use the notation $\mathbf{m}$ in Algorithm 1 as well as in the text to refer to these meta-graph features used by METAGL.

### G.2  META-LEARNING OBJECTIVE FOR SPARSE PERFORMANCE MATRIX

Given a sparse performance matrix $\mathbf{P}$, meta-training of METAGL can be performed by modifying the top-1 probability (Equation (3)) and the loss function (Equation (4)), such that the missing entries in $\mathbf{P}$ are ignored as follows:

$$p_{\text{top1}}^{\widehat{\mathbf{P}}_i}(j) = \frac{\mathbf{I}_{p_{ij}}(\pi(\widehat{p}_{ij}))}{\sum_{k=1}^m \mathbf{I}_{p_{ik}}(\pi(\widehat{p}_{ik}))} = \frac{\mathbf{I}_{p_{ij}}(\exp(\widehat{p}_{ij}))}{\sum_{k=1}^m \mathbf{I}_{p_{ik}}(\exp(\widehat{p}_{ik}))}, \tag{7}$$

$$L(\mathbf{P}, \widehat{\mathbf{P}}) = -\sum_{i=1}^n \sum_{j=1}^m \mathbf{I}_{p_{ij}} \left( p_{\text{top1}}^{\mathbf{P}_i}(j) \log \left( p_{\text{top1}}^{\widehat{\mathbf{P}}_i}(j) \right) \right). \tag{8}$$

where $\mathbf{I}_{p_{ij}}(\cdot)$ is defined as

$$\mathbf{I}_{p_{ij}}(x) = \begin{cases} x & \text{if } p_{ij} \text{ exists in the observed performance matrix } \mathbf{P}, \\ 0 & \text{if } p_{ij} \text{ is missing in the observed performance matrix } \mathbf{P}. \end{cases} \tag{9}$$

Thus the supervision signal for each graph comes only from the model performances observed on it. If an entire row in $\mathbf{P}$ is empty, the loss terms for the corresponding graph are dropped from Equation (8).

### G.3  G-M NETWORK

Figure 7 illustrates the G-M network (graph-model network) (Section 4.4), which is a multi-relational bipartite network between graph nodes and model nodes. In the G-M network, model and graph nodes are connected via five types of edges (*e.g.*, P-m2m, P-g2m, M-g2g), which is shown as edges with distinct line styles and colors. Note that while Figure 7 shows only one edge per edge type, in the G-M network, each node is connected to its top-$k$ similar nodes.

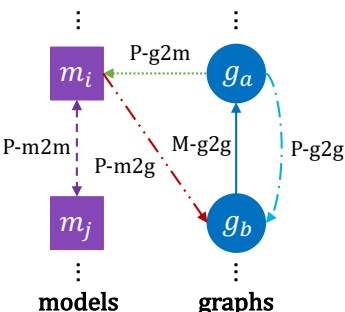

Figure 7: An illustration of the G-M network (Section 4.4), which is a multi-relational bipartite network between model nodes and graph nodes that are connected via five types of edges (*e.g.*, P-m2m, M-g2g). Note that only a subset of the edges in the G-M network is shown here for illustration purposes. See Section 4.4 for more details.

### G.4 ATTENTIVE GRAPH NEURAL NETWORKS AND HETEROGENEOUS GRAPH TRANSFORMER

The embedding function $f(\cdot)$ in METAGL (Section 4.4) produces embeddings of models and graphs via weighted neighborhood aggregation over the multi-relational G-M network. Specifically, we define $f(\cdot)$ using Heterogeneous Graph Transformer (HGT) (Hu et al., 2020), which is a relation-aware graph neural network (GNN) that performs attentive neighborhood aggregation over the G-M network. Let $\mathbf{z}_t^\ell$ denote the node $t$'s embedding produced by the $\ell$-th HGT layer, which becomes the

---

**Algorithm 1:** METAGL: Offline Meta-Training (Top) and Online Model Selection (Bottom)

---

**Input:** Meta-train graph database $\mathcal{G}$, model set $\mathcal{M}$, embedding dimension $k$
**Output:** Meta-learner for model selection
/* (Offline) Meta-Learner Training (Sec. 4.1)                                              */
1 Train & evaluate models in $\mathcal{M}$ on graphs in $\mathcal{G}$ to get performance matrix $\mathbf{P}$
2 Extract meta-graph features $\mathbf{M}$ for each graph $G_i$ in $\mathcal{G}$ (Sec. 4.3)
3 Factorize $\mathbf{P}$ to obtain latent graph factors $\mathbf{U}$ and model factors $\mathbf{V}$, *i.e.*, $\mathbf{P} \approx \mathbf{U}\mathbf{V}^\mathsf{T}$
4 Learn an estimator $\phi(\cdot)$ such that $\phi(\mathbf{m}) = \widehat{\mathbf{U}}_i \approx \mathbf{U}_i$
5 Create meta-train graph $\mathcal{G}_{\text{train}}$ (Sec. 4.4)
6 **while** *not converged*
7     **for** $i = 1, \ldots, n$ **do**
8         Get embeddings $f(\mathbf{W}[\mathbf{m}; \phi(\mathbf{m})])$ of train graph $G_i$ on $\mathcal{G}_{\text{train}}$
9         **for** $j = 1, \ldots, m$ **do**
10             Get embeddings $f(\mathbf{V}_j)$ of each model $M_j$ on $\mathcal{G}_{\text{train}}$
11             Estimate $\widehat{p}_{ij} = \langle f(\mathbf{W}[\mathbf{m}; \phi(\mathbf{m})]), f(\mathbf{V}_j) \rangle$ (Eqn. 2)
12         **end**
13     **end**
14     Compute meta-training loss $L(\mathbf{P}, \widehat{\mathbf{P}})$ (Eqn. 4) and optimize parameters
15 **end**

---

**Input:** new graph $G_{\text{test}}$
**Output:** selected model $M^*$ for $G_{\text{test}}$
/* (Online) Model Selection (Sec. 4.2)                                                     */
16 Extract meta-graph features $\mathbf{m}_{\text{test}} = \psi(G_{\text{test}})$
17 Estimate latent factor $\widehat{\mathbf{U}}_{\text{test}} = \phi(\mathbf{m}_{\text{test}})$ for test graph $G_{\text{test}}$
18 Create the test G-M network $\mathcal{G}_{\text{test}}$ by extending $\mathcal{G}_{\text{train}}$ with new edges between test graph node and existing nodes in $\mathcal{G}_{\text{train}}$ (Sec. 4.4)
19 Get embeddings $f(\mathbf{W}[\mathbf{m}_{\text{test}}; \widehat{\mathbf{U}}_{\text{test}}])$ of test graph on $\mathcal{G}_{\text{test}}$
20 Get embeddings $f(\mathbf{V}_j)$ of each model $M_j$ on $\mathcal{G}_{\text{test}}$
21 Return the best model $M^* \leftarrow \arg\max_{M_j \in \mathcal{M}} \langle f(\mathbf{W}[\mathbf{m}_{\text{test}}; \widehat{\mathbf{U}}_{\text{test}}]), f(\mathbf{V}_j) \rangle$

---

input of the $(\ell+1)$-th layer. Given $L$ total layers, the final embedding $\mathbf{h}_t$ of node $t$ is obtained to be the output from the last layer, *i.e.*, $\mathbf{z}_t^L$. In general, node embeddings $\mathbf{z}_t^\ell$ produced by the $\ell$-th layer in an attention-based GNN, such as HGT, can be expressed as:

$$\mathbf{z}_t^\ell = \underset{\forall s \in N(t), \forall e \in E(s,t)}{\textbf{Aggregate}} \Big( \textbf{Attention}(s,t) \cdot \textbf{Message}(s) \Big) \tag{10}$$

where $s$ and $t$ are source and target nodes, respectively; $N(t)$ denotes all the source nodes of node $t$; and $E(s,t)$ denotes all edges from node $s$ to $t$. There are three basic operators: **Attention**, which assigns different weights to neighbors based on the estimated importance of node $s$ with respect to target node $t$; **Message**, which extracts the message vector from the source node $s$; and **Aggregate**, which aggregates the neighborhood messages by the attention weight.

HGT effectively processes multi-relational graphs, such as the proposed G-M network, by designing all of the above three operators to be aware of node types and edge types, *e.g.*, by employing distinct set of projection weights for each type of nodes and edges, and utilizing node- and edge-type dependent attention mechanisms. We refer the reader to (Hu et al., 2020) for the details of how HGT defines the above three operators. In summary, METAGL computes the embedding function $f(\mathbf{x}_t)$ by providing node $t$'s input features $\mathbf{x}_t$ as the initial embedding (*i.e.*, $\mathbf{z}_t^0$) to HGT, and returning $\mathbf{z}_t^L$, the output from the last layer, which is computed over the G-M network via relation-aware attentive neighborhood aggregation.

### G.5 TIME COMPLEXITY ANALYSIS

We now state the time complexity of our approach for inferring the best model given a new unseen graph $G' = (V', E')$. Let $\mathcal{G} = (\mathcal{V}, \mathcal{E})$ be the G-M network, which is comprised of model nodes and graph nodes, and induced by $c$-NN (nearest neighbor) search (Section 4.4 and Appendix G.3). Let $k$ denote the embedding size, and $h$ be the number of attention heads in HGT (Appendix G.4). The time complexity of METAGL is

$$\mathcal{O}(q|E'|\Delta + |\mathcal{V}|ck^2/h) \tag{11}$$

where $q$ is the number of meta-graph feature extractors, and $|E'|$ is the number of edges in the new unseen graph $G'$. Note that both $q$ and $\Delta$ are small and thus negligible. Hence, METAGL is fast and efficient.

*Meta-Graph Feature Extraction:* The first term of the above time complexity includes the time required to estimate the frequency of all network motifs with $\{2,3,4\}$-nodes, which is $\mathcal{O}(|E'|\Delta)$ in the worst case where $\Delta$ is a small constant representing the maximum sampled degree which can be set by the user. See Ahmed et al. (2016) for further details. The other structural meta-feature extractors such as PageRank all take at most $\mathcal{O}(|E'|)$ time. Furthermore, our approach is flexible and supports any set of meta-graph feature extractors. Thus, it is straightforward to see that we can achieve a slightly better time by restricting the set of such meta-graph feature extractors to those that can be computed in time that is linear in the number of edges of any arbitrary graph. Hence, in this case, the $\Delta$ term is dropped and we have simply $\mathcal{O}(q|E'| + |\mathcal{V}|ck^2/h)$. Also, note that feature extractors are independent of each other, and thus can be run in parallel.

*Embedding Models and Graphs:* To augment the G-M network $\mathcal{G}$ given a new graph $G'$, we find a fixed number of nearest neighbors for $G'$, which takes $\mathcal{O}(|\mathcal{V}|k)$ time. Then we embed models and graphs by applying HGT over the G-M network $\mathcal{G} = (\mathcal{V}, \mathcal{E})$. Assuming an HGT with $h$ attention heads, the time to employ HGT over $\mathcal{G}$ is $\mathcal{O}(|\mathcal{V}|k^2 + |\mathcal{E}|k^2/h)$. More specifically, the time taken for Attention($\cdot$) and Message($\cdot$) functions is $\mathcal{O}(|\mathcal{V}|k^2 + |\mathcal{E}|k^2/h)$, where $\mathcal{O}(|\mathcal{V}|k^2)$ is for feature transformation by $h$ heads for all nodes, and $\mathcal{O}(|\mathcal{E}|k^2/h)$ is for message transformation/attention computation over each edge. Similarly, Aggregate($\cdot$) step takes $\mathcal{O}(|\mathcal{V}|k^2 + |\mathcal{E}|k)$ time. Assuming $k^2/h > k$, the time for Aggregate($\cdot$) can be absorbed into the time for other steps. Further, given that the G-M network is induced by $c$-NN search, we have that $\mathcal{O}(|\mathcal{E}|) = \mathcal{O}(c|\mathcal{V}|)$, and thus the time complexity for embedding models and graphs is $\mathcal{O}(|\mathcal{V}|ck^2/h)$.

## H ADDITIONAL RELATED WORK

### H.1 MODEL SELECTION IN MACHINE LEARNING

In this section, we provide a further review of model selection in machine learning, which we group into two categories.

*Evaluation-Based Model Selection*: A majority of model selection methods belong to this category. Representative techniques used by these methods include grid search (Liashchynskyi & Liashchynskyi, 2019), random search (Bergstra & Bengio, 2012), early stopping-based (Golovin et al., 2017) and bandit-based (Li et al., 2017b) approaches, and Bayesian optimization (BO) (Snoek et al., 2012; Wu et al., 2019b; Falkner et al., 2018). Among them, BO methods are more efficient than grid or random search, requiring fewer evaluations of hyperparameter configurations (HCs), as they determine which HC to try next in a guided manner using prior experience from previous trials. Since these methods perform model training or evaluation multiple times using different HCs, they are much less efficient than the following group of methods.

*Evaluation-Free Model Selection*: Methods in this category do not require model evaluation for model selection. A simple approach (Abdulrahman et al., 2018) identifies the best model by considering the models' rankings observed on prior datasets. Instead of finding the globally best model, ISAC (Kadioglu et al., 2010) and AS (Nikolić et al., 2013) select a model that performed well on similar datasets, where the dataset similarity is modeled in the meta-feature space via clustering (Kadioglu et al., 2010) or $k$-nearest neighbor search (Nikolić et al., 2013). A different group of methods perform optimization-based model selection, where the model performance is estimated by modeling the relation between meta-features and model performances. Supervised Surrogates (Xu et al., 2012) learns a surrogate model that maps meta-features to model performance. Recently, MetaOD (Zhao et al., 2021) outperformed all of these methods in selecting outlier detection algorithms. As a method in this category, the proposed METAGL builds upon MetaOD and extends it for an effective and automatic GL model selection. Most importantly, METAGL selects a *graph learning* model (*e.g.*, link predictor) for the given *graph*, while MetaOD selects an *outlier detection* (OD) model for the given dataset ($n$-dimensional input features). To this end, METAGL designs meta features to capture the characteristics of graphs, while MetaOD designs meta features specialized for OD tasks. Also, they adopt different meta-training objectives: METAGL adapts the top-1 probability for meta-training, whereas MetaOD uses an NDCG-based objective. Furthermore, METAGL learns the embeddings of models and graphs by applying a heterogeneous GNN over the G-M network, which allows a flexible modeling of the relations between different models and graphs. By contrast, in MetaOD, the embeddings of models and datasets are optimized separately, where the relations between models and datasets are modeled rather indirectly via reconstructing the performance matrix. Note that all of these earlier methods, except the first simple approach, rely on meta-features, and they focus on non-graph datasets. By using the proposed meta-graph features, they could be applied to the graph learning model selection task.

## H.2 COMPARISON WITH MODEL-AGNOSTIC META-LEARNING (MAML) (FINN ET AL., 2017)

MAML employs meta-learning to train a model's initial parameters such that the model can perform well on a new task after the parameters have been updated via a few gradient steps using the data from the new task. In other words, given a model, MAML initializes one specific model's parameters via meta-learning over multiple observed tasks, such that the meta-trained model can quickly adapt to a new task after learning from a small number of new data (*i.e.*, few-shot learning). On the other hand, METAGL employs meta-learning to carry over the prior knowledge of multiple different models' performance on different graphs for evaluation-free selection of graph learning algorithms.

Since MAML meta-trains a specific model for fast adaptation to a new dataset, it is not for selecting a model from a model set consisting of a wide variety of learning algorithms. Further, MAML fine-tunes a meta-trained model in a few-shot learning setup, whereas in our problem setup, no training and evaluation is to be done given a new graph dataset. Due to these reasons, MAML is not applicable to the proposed evaluation-free GL model selection problem (Problem 1).

## I ADDITIONAL RESULTS

### I.1 EFFECTIVENESS OF META-GRAPH FEATURES

Figure 8 shows how accurately meta-learners can perform model selection when they use the proposed meta-graph features (Section 4.3) vs. six state-of-the-art graph-level embedding (GLE) techniques, *i.e.*, GL2Vec (Chen & Koga, 2019), Graph2Vec (Narayanan et al., 2017), GraphLoG (Xu et al., 2021), WaveletCharacteristic (Wang et al., 2021), SF (de Lara & Pineau, 2018), and LDP (Cai & Wang, 2019). As discussed in Section 5.3, all meta-learners achieve a higher model selection accuracy nearly

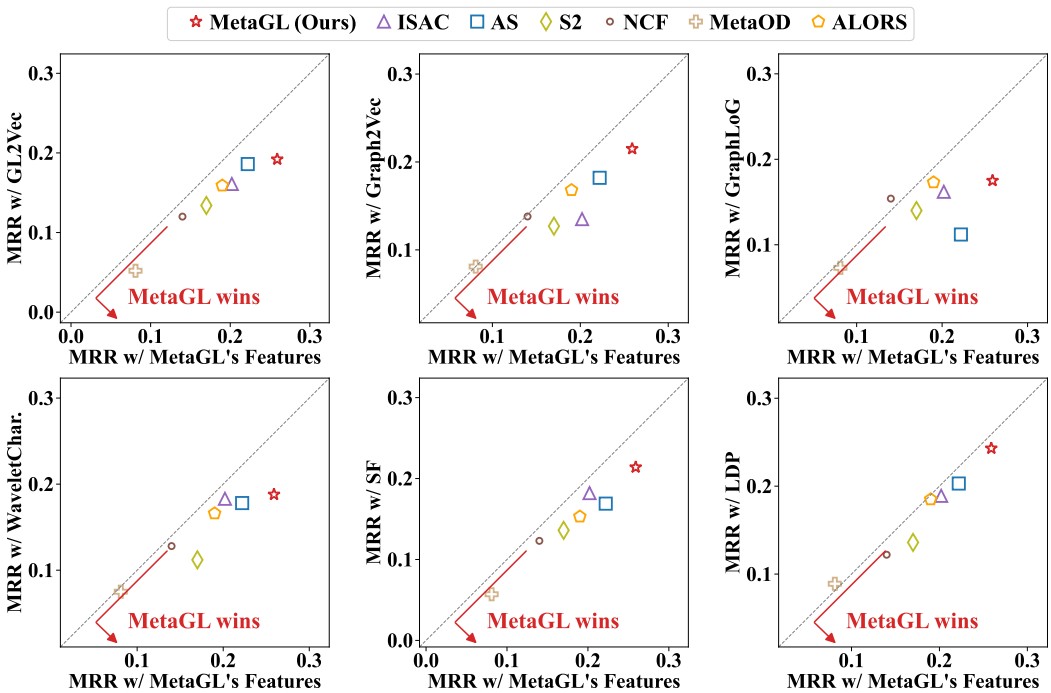

Figure 8: Using the proposed meta-graph features (Section 4.3), all meta-learners nearly consistently perform better than when they use existing graph-level embedding methods (most of the points are below the diagonal).

consistently by using METAGL's meta-graph features than when they use these GLE techniques, and METAGL outperform all meta-learners, regardless of which features are used.

Table 7: METAGL is fast and incurs negligible overhead. The runtime (in secs) of naive model selection (*i.e.*, training each method using all configurations in the model set $\mathcal{M}$), versus the runtime of METAGL, *i.e.*, the time to generate meta-graph features (the penultimate row) plus the time taken for model prediction on average (the last row). Datasets are taken from (Rossi & Ahmed, 2015).

| | | soc-wiki-Vote | ia-reality | tech-routers-rf | ca-cora | power-US-Grid | web-EPA | socfb-Caltech | tech-pgp |
|---|---|---|---|---|---|---|---|---|---|
| **line** | | 5.45 | 5.85 | 5.40 | 6.47 | 8.23 | 7.28 | 8.19 | 10.15 |
| **node2vec** | | 65.28 | 504.38 | 154.54 | 159.32 | 315.35 | 317.55 | 184.09 | 722.66 |
| **deepwalk** | | 7.03 | 55.01 | 16.89 | 17.95 | 35.62 | 33.09 | 18.24 | 84.35 |
| **HONE** | | 203.71 | 53.15 | 552.31 | 276.11 | 127.4 | 882.35 | 737.49 | 2082.06 |
| **node2bits** | | 64.85 | 113.06 | 92.22 | 93.37 | 211.3 | 117.55 | 106.66 | 284.42 |
| **deepGL** | | 145.93 | 633.44 | 331.87 | 504.2 | 1027.25 | 880.03 | 445.89 | 2349.3 |
| **GraphSage** | | 272.97 | 1451.87 | 513.18 | 283.35 | 115.86 | 1020.30 | 2586.93 | 1171.54 |
| **GCN** | | 26.30 | 57.10 | 45.64 | 52.96 | 57.38 | 66.12 | 94.65 | 163.4 |
| **METAGL** | (meta feat. gen.) | 0.16 | 0.97 | 0.36 | 0.4 | 1.07 | 0.78 | 0.61 | 2.05 |
| | (model pred.) | | | | 0.39 | | | | |

## I.2 MODEL SELECTION TIME

Table 7 shows results comparing the runtime (in seconds) for naive model selection with the runtime of METAGL. Note that naive model selection requires training and evaluating each method in the model set, while in METAGL, the runtime involves only the time to generate meta-graph feature (penultimate row) and predict the best model via a forward pass (last row). Results show that METAGL is fast, and incurs negligible computational overhead.

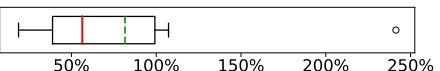

Figure 9: Distribution of the time for METAGL to make a prediction / the time to create meta-graph features (in percentage). Red and green lines denote the median and mean, respectively.

Figure 9 shows the distribution of the time taken for METAGL to predict the best model, divided by the time taken by METAGL for creating meta-graph features (in percentage). On average, it takes ~20% less time for METAGL to make an inference than to generate meta-graph features.

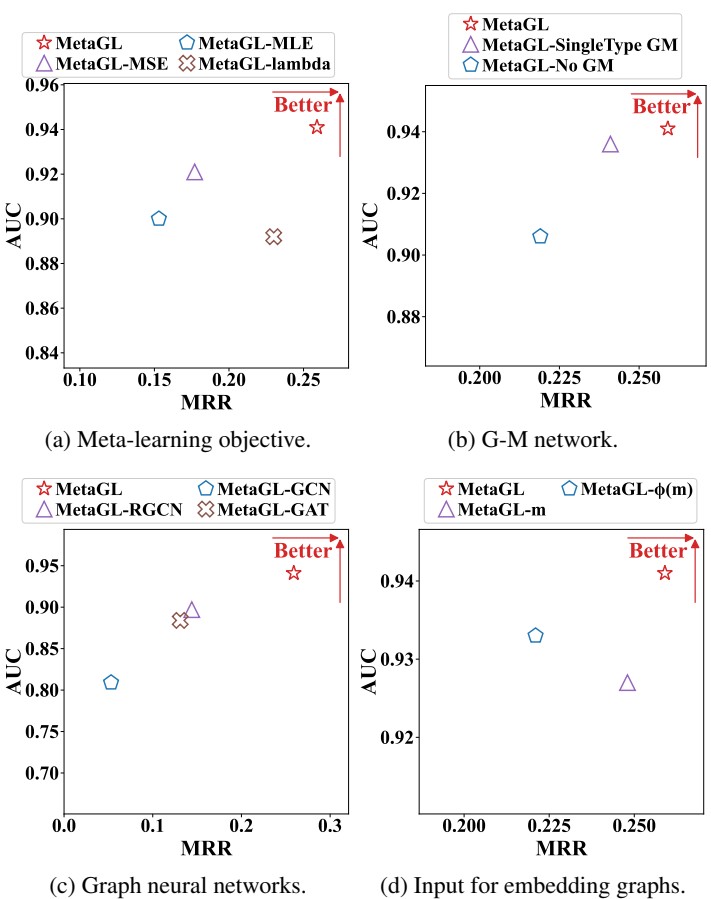

(a) Meta-learning objective.
(b) G-M network.

(c) Graph neural networks.
(d) Input for embedding graphs.

Figure 10: Ablation study of METAGL on several components that METAGL uses for meta-learning. The proposed METAGL achieves the best performance, in comparison to the variants that adopt different modeling choices.

## I.3 ABLATION STUDY

Figure 10 presents ablation studies on several components of METAGL for meta-learning, namely, meta-learning objective, the G-M network, the GNN encoder, and the input for embedding graphs. To that end, we compare METAGL against several variants that differ from METAGL in just one aspect, which we explain below. In Figure 10, METAGL refers to the proposed meta-learner as described in Section 4.

### I.3.1 META-LEARNING OBJECTIVE

To optimize the framework such that it can find the best model, METAGL employs the top-1 probability as its meta-learning objective (see Section 4.1). We evaluate how METAGL's performance changes when it uses different loss functions for meta-training. **METAGL-MSE** uses mean squared error (MSE), **METAGL-MLE** employs ListMLE (Xia et al., 2008), and **METAGL-lambda** uses LambdaLoss (Wang et al., 2018). Figure 10a shows that METAGL with the top-1 probability most effectively identifies the top-performing model for the new graph, while variants that employ different learning objectives obtain suboptimal results in terms of both MRR and AUC.

### I.3.2 G-M Network

To learn the embeddings of models and graphs, METAGL uses a G-M network (GM), which is a multi-relational network containing multiple types of nodes and edges (see Section 4.4 and Appendix G.3). **METAGL-SingleType GM** is a variant of METAGL that uses a single-relational G-M network, where all of the nodes and edges in the original G-M network have been converted to be of the same type. **METAGL-No GM** is another variant where METAGL does not use the G-M network at all; instead, the input features of models and graphs, which are normally provided as input to the GNN in METAGL, were directly taken to be the final model and graph embeddings. Figure 10b shows that METAGL with the multi-relational G-M network is the most effective, and in utilizing the G-M network, it is helpful to be able to distinguish between the multiple types of nodes and edges. Further, in comparison to METAGL-No GM, learning the embeddings of graphs and models via the G-M network greatly improves the model selection performance.

### I.3.3 Graph Neural Networks

To effectively learn the embeddings of models and graphs over a multi-relational G-M network, METAGL utilizes an attentive, heterogeneous graph neural network (GNN) as its graph encoder (see Section 4.4 and appendix G.4), which can adaptively determine the aggregation weight of the neighbors (thus *attentive*), while being aware of and utilizing the multiple node and edge types in learning the embeddings (hence *heterogeneous*). To see how helpful attentive, heterogeneous GNNs are in learning effective model and graph embeddings, here we create the following variants of METAGL, which use GNNs that are not attentive and/or not heterogeneous (*i.e.*, no mechanism to handle different node and edge types). **METAGL-RGCN** uses RGCN (Schlichtkrull et al., 2018), which is a heterogeneous GNN, but is not attentive. **METAGL-GAT** uses GAT (Velickovic et al., 2018), which is an attentive GNN, but is not heterogeneous. **METAGL-GCN** uses GCN Kipf & Welling (2017), which is neither heterogeneous nor attentive. In Figure 10c, the highest performance is achieved by METAGL that uses heterogeneous and attentive GNNs, while the lowest performance is obtained when it uses GCN, which is not attentive nor heterogeneous. Both the ability to perform attentive neighborhood aggregation, and to be able to distinguish between and utilize different node and edge types are essential in effectively capturing the complex relations between graphs and models.

### I.3.4 Input for Embedding Graphs

We have two sources of information for graphs: one is the performance matrix $\mathbf{P}$, and the other is graph data. Meta-graph feature $\mathbf{m}$ captures the information from the graph data, while $\phi(\cdot)$ captures the information from the performance matrix by estimating the latent graph factor obtained by factorizing $\mathbf{P}$. Therefore, the two terms $\mathbf{m}$ and $\phi(\mathbf{m})$ in Equation (2) aim to incorporate two different sources of information. Here we create the following two variants of METAGL: In Equation (2), **METAGL-$\mathbf{m}$** uses $\mathbf{m}$ alone, and similarly **METAGL-$\phi(\mathbf{m})$** uses only $\phi(\mathbf{m})$. Figure 10d shows that they provide complementary information, and using both leads to the best result.

### I.4 Sensitivity to the Variance of the Performance Matrix

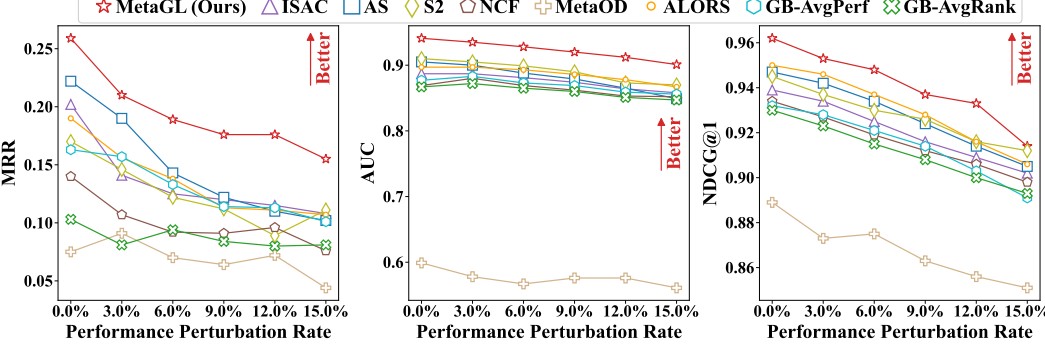

Figure 11: Across varying performance perturbation rates, METAGL consistently improves upon existing model selection approaches, achieving up to 53% higher MRR than the best baseline.

We evaluate how sensitive METAGL and baselines are to the variance of the performance matrix $\mathbf{P}$, which can be introduced due to the non-deterministic nature of model training and evaluation. For evaluation, we perturb all entries of the fully-observed performance matrix $\mathbf{P}$ to varying degrees, and measure the model selection performance using the perturbed performance matrix $\widetilde{\mathbf{P}}$. Specifically, given a performance perturbation rate $r$ ($r \geq 0$), we replace each performance entry $n$ in $\mathbf{P}$ with the perturbed performance $\widetilde{n}$, which is uniformly randomly sampled from $[n(1 - r/2), n(1 + r/2)]$, bounded by 0 and 1. For instance, given a perturbation rate of $0.2$, a mean average precision of $0.5$ is perturbed by sampling from $[0.45, 0.55]$. As a result, higher perturbation rate makes the perturbed matrix more random, and thus harder to predict. Figure 11 shows that while an increased perturbation rate indeed makes model selection harder for all methods, METAGL consistently outperforms baseline model selection approaches across varying performance perturbation rates, obtaining up to 53% higher MRR than the best baseline.

## I.5 COMPARISON WITH THE ACTUAL BEST PERFORMANCE

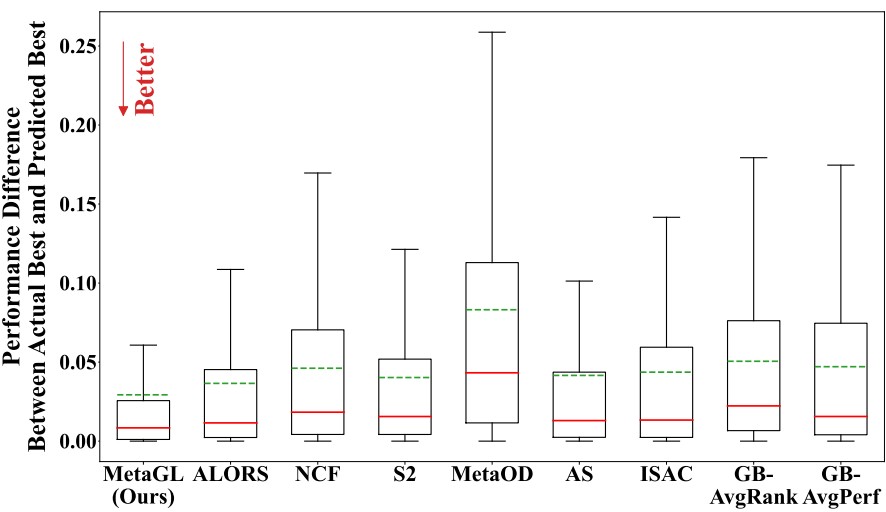

Figure 12: The smallest performance difference between the actual best and predicted best models is obtained with METAGL (27% and 20% less than the second smallest difference obtained by ALORS in terms of median and mean, respectively). METAGL's distribution is also much tighter than others.

Here we compare the actual best performance observed for each graph against the performance of the model chosen by different model selection approaches. Figure 12 is a box plot showing how those performance differences are distributed. Figure 12 shows that using METAGL leads to the smallest performance difference between the actual best and the predicted best models (27% and 20% less than the second smallest difference obtained by ALORS in terms of median and mean, respectively). Further, the performance difference distribution obtained with METAGL is much tighter than that of baselines.

