# OpenReview forum: "MetaGL: Evaluation-Free Selection of Graph Learning Models via Meta-Learning"
_ICLR.cc/2023/Conference — ICLR 2023 poster_

### Official Review · Reviewer_P6vP · 2022-10-23

**Confidence:** 3
**Correctness:** 3
**Technical Novelty And Significance:** 3
**Empirical Novelty And Significance:** 3
**Recommendation:** 6

**Clarity, Quality, Novelty And Reproducibility:**

For the Clarity:
1. There are several typos and grammar errors in this paper. For example, “smarter, more efficient” should be “smarter and more efficient”. “is fully automatic it does not require” lacks the word “since”. “number of wedges” should be “number of edges”.

2. The paragraph starting with “Benchmark Data/Code:” appears twice in the main paper, which is redundant.

For the Quality and Novelty:
The quality seems to be solid and the studied problem is novel. The specific modules in the framework are less novel, while the framework is well designed.

For the Reproducibility:
Following the above, the authors claim that they release the code and datasets, which are not found in the paper. No links are provided.


**Strength And Weaknesses:**

Strengths:
1. The framework solves the novel problem of evaluation-free model selection in graph learning, which is crucial and challenging in real-world scenarios.
2. The experimental results are comprehensive and adequate. The authors also provide additional results regarding the effectiveness of each module and the efficiency of the framework.
3. The paper is well organized and easy to follow.




Weaknesses:

1. The structural information in each graph seems to be only incorporated via the meta-graph features. However, this can also be achieved by a specific GNN with a readout function. That being said, the improvement of the proposed framework is hardly believed to result from the incorporation of structural information.
2. The input for computing the latent embedding of a graph to function f() is not rational. If φ(m) is solely dependent on m, why the input of f() is a combination of m and φ(m), instead of only m or φ(m)? In this way, the information of m can also be preserved. On the other hand, combining both will seem redundant.
3. The experimental part only includes the link prediction task. Although this task is important in graph mining, the proposed method should be capable of various graph mining tasks (considering that the scenario will generally be applications). However, the lack of evaluation on other graph mining tasks will cause the paper to be less convincing.
4. The paper lacks sufficient theoretical analysis to support the claim of efficacy and efficiency. The empirical results are plentiful, while further analysis would be beneficial.


**Summary Of The Paper:**

This paper proposes a novel framework to conduct evaluation-free model selection for graph learning models, i.e., without having to train/evaluate any model on the new graph. The framework learns latent embeddings for observed models and the corresponding performance on observed graphs. Moreover, meta-graph features are computed for each graph based on specific structures so that the graph similarity can be measured. The extensive experiments demonstrate the effectiveness of the proposed framework.

**Summary Of The Review:**

The paper solves a novel and crucial problem of evaluation-free model selection for graph learning. The experimental results are comprehensive. However, the major drawback lies in the lack of other graph mining tasks except for link prediction. The design is somewhat novel while also lacking theoretical analysis.

---

> ### Author Response · Authors · 2022-11-19
> **Response to Reviewer P6vP**
>
> Thank you for your thoughtful comments. Please find our response to your comments and questions below. We have also updated and highlighted the paper in light of your comment, which is marked in blue in the revised paper.
>
> > Q1. “The structural information in each graph seems to be only incorporated via the meta-graph features. However, this can also be achieved by a specific GNN with a readout function. That being said, the improvement of the proposed framework is hardly believed to result from the incorporation of structural information.”
>
> In our evaluation (Sec 5.3 and Appendix I.1) of the proposed meta-graph features, and existing graph embedding approaches that can incorporate structural information (such as GNNs with a readout function), meta-learners perform better when they use the meta-graph features in nearly all cases (most points are below the diagonal in Figures 5 and 8). This result indicates that the quality of structural information does affect the efficacy of meta-learning-based model selection.
>
> Importantly, in other experiments, all meta-learners were provided with the same meta-graph features as input. Therefore, in these cases, the performance differences between MetaGL and baselines stem from the differences in the way they process the same input information (i.e., structural information in the meta-graph features as well as the prior model performances) for effective knowledge transfer. Experimental results show that MetaGL improves upon baselines, when they are given the same information.
>
> > Q2. “The input for computing the latent embedding of a graph to function f() is not rational. If φ(m) is solely dependent on m, why the input of f() is a combination of m and φ(m), instead of only m or φ(m)? In this way, the information of m can also be preserved. On the other hand, combining both will seem redundant.”
>
> In our problem setting, we have two sources of information for graphs: one is the performance matrix, and the other is graph data. Meta-graph feature m captures the information from the graph data, while φ() captures the information from the performance matrix by estimating the latent graph factor U obtained by factorizing the performance matrix P. Therefore, although φ() is dependent on m, the two terms m and φ(m) aim to incorporate two different sources of information. The following results obtained by using m with φ(m), or only either m or φ(m) show that they provide complementary information, and using both leads to the best result.
> - m+φ(m): MRR (0.261), AUC(0.941), NDCG@1(0.962)
> - φ(m): MRR (0.221), AUC(0.933), NDCG@1(0.962)
> - m: MRR (0.248), AUC(0.927), NDCG@1(0.952)
>
> We included this result in the ablation study (Appendix I.3.4).
>
> > Q3. “The experimental part only includes the link prediction task.”
>
> In this first study on evaluation-free GL model selection, we investigated link prediction since it is a fundamental and unique task for graph-structured data, with many important applications like recommendation and knowledge reasoning. Also, one can easily cast other problems such as node classification as a link prediction problem (e.g., Fadaee & Haeri (2019)). Also, the proposed benchmark dataset could be constructed through a time and computation expensive process, which is currently the largest one available, with performances obtained via 127,323 model training and evaluations (301 graphs x 423 models). In light of these considerations, we focused on link prediction in this paper.
>
> > Q4. “The paper lacks sufficient theoretical analysis to support the claim of efficacy and efficiency. The empirical results are plentiful, while further analysis would be beneficial.”
>
> Thank you for the suggestion. We have now included a further discussion and analysis of MetaGL in Appendix G.5.
>
> [Clarity]
> 1. We fixed these typos and grammar errors. Thank you for pointing out those mistakes. Number of wedges (a path of length 2) is indeed the feature that we used in this work. We have clarified this in the paper as well.
> 2. We removed the paragraph from the Conclusion. It now appears once at the end of the Introduction.
>
> [Reproducibility] We will release the code and benchmark datasets when the paper gets published.

---

> > ### Comment · Reviewer_P6vP · 2022-11-21
> > **Response**
> >
> > Thank you for your response. It is comprehensive and helped me understand your design better.

---

### Official Review · Reviewer_3hVZ · 2022-10-24

**Confidence:** 5
**Clarity, Quality, Novelty And Reproducibility:** The paper is clearly written and orig…
**Correctness:** 3
**Technical Novelty And Significance:** 3
**Empirical Novelty And Significance:** 3
**Recommendation:** 6

**Strength And Weaknesses:**

### Strengths:
1. This paper first considers how to select an effective GL link prediction model for a new graph without searching process. It is a new and meaningful problem.

2. The experiments are solid and can support the contributions: effectiveness of the meta-learning framework and global statistical features.

3. The construction of G-M Network makes sense and further improves the performance.

### Weaknesses:
1. It seems that the extracting of some meta-features (e.g., triangles and Page Rank score) is time-consuming which limits the application on the large-scale graph.

2. The GRL methods in the model set are limited and the model set may not contain some new GRL methods in recent years.

3. It seems to be missing the training details and metrics of the link prediction subproblem.


**Summary Of The Paper:**

This paper considers the problem of how to select an effective GL model for link prediction task on a new graph without training/evaluating process. To deal with this problem, the authors propose a meta-learning based matrix factorization (MF) approach MetaGL. MetaGL adopt the meta graph features to represent graphs, which can convert the transductive MF to an inductive manner for a new graph.
The proposed method is evaluated on 301 graphs via 5-fold cross validation with 412 candidate models.


**Summary Of The Review:**

The paper considers a new problem of selecting effective GL models in an evaluation-free manner and proposes an effective approach to handle this problem with solid experiments. However, there are some issues with the scalability and limited GRL methods in the model set.

---

> ### Author Response · Authors · 2022-11-19
> **Response to Reviewer 3hVZ**
>
> We appreciate your thoughtful comments. Please find our response to your comments and questions below. We have also updated and highlighted the paper in light of your comment, which is marked in green in the revised paper.
>
> > Q1. “It seems that the extracting of some meta-features (e.g., triangles and Page Rank score) is time-consuming which limits the application on the large-scale graph.”
>
> Thank you for this feedback. We have now included a discussion of this in Appendix G.5. To clarify, our approach is flexible and can leverage any set of structural meta-features. Nevertheless, PageRank is linear in the number of edges and triangles can be computed in nearly linear time and estimated even faster.
>
> > Q2. “The GRL methods in the model set are limited and the model set may not contain some new GRL methods in recent years.”
>
> To the best of our knowledge, the proposed benchmark dataset is the largest one available, which contains performances for 127,323 model training and evaluations (423 models x 301 graphs from various domains). We will make the code and benchmark datasets publicly available when the paper gets published, and new graphs and models can easily be added to it. Furthermore, the state-of-the-art GL model on one dataset may have poor performance on others. In light of this, our goal in creating a benchmark dataset has been to facilitate research and evaluation of different model selection approaches, which does not depend on the inclusion of latest models.
>
> > Q3. “It seems to be missing the training details and metrics of the link prediction subproblem.”
>
> Thank you for pointing this out. We have added the following in Appendix F.1, which provides training details and metrics of the link prediction subproblem.
>
> Given a graph G, we first hold out 10% of the edges in graph G to be used for evaluation, and train GL models with the resulting subgraph for link prediction. The training of GL models was performed by sampling 20 negative edges per positive edge, estimating the link score by applying a dot product between the two corresponding node embeddings, followed by a sigmoid function, and then optimizing a binary cross entropy loss for the positive and negative edge scores. For evaluation, we randomly sampled the same number of negative edges as the positive edges, and evaluated the estimated link scores in terms of mean average precision.

---

### Official Review · Reviewer_uUz4 · 2022-10-26

**Confidence:** 4
**Correctness:** 3
**Technical Novelty And Significance:** 3
**Empirical Novelty And Significance:** 2
**Recommendation:** 5

**Clarity, Quality, Novelty And Reproducibility:**

The presentation is overall clear enough the catch the main idea. However, I have concerns on the novelty of the proposed method.

**Strength And Weaknesses:**

Strength:
1. The presentation and writing are good enough for clearly presenting the motivation and corresponding solution.
2. Meta-features for graph data are clearly defined by fully extracting the graph structural knowledge.
3. Extensive experiments are conducted to demonstrate the superiority of proposed method to state-of-the-art meta model selection method.

Weakness:
1. The novelty is limited to applying the principles of MetaOD to graph data by matching meta-features of graph data with model representations. It's interesting to compare the way to run "meta-learning" in this work with the optimization-based meta-learning methods like MAML [1]. The common thing shared by them is that finally selected model should have impressive prediction performance. The principles emphasized in this work pays attention to select the best one from the stored models which are trained on historical data. While optimization-based meta-learning can capture the unique knowledge existing in the input data just by fine-tuning models. I strongly suggest to have a discussion on the difference between them and make a comparison over their prediction performance.

2. Though the proposed method can fast search one model from the candidates, it's still difficult to believe that only comparing the meta-features extracted from graph structures can select the exact model for the input data. What if the extracted meta-feature can not represent the node features distribution differences existing in the graphs? Is it still the best choice for model selection? or is the learning-free model selection suitable to deal with the knowledge gap between the new graph and those graphs in the data pool?

3. Last but not least, I still have concern on the generalization of the proposed method. It's well known that graph consists of unique node sets. How is possible that the models trained on different graphs can be applied to a graph with totally different nodes? Maybe authors should present some concrete applications that are suitable to be applied the proposed method.


References:
1. Finn, Chelsea, Pieter Abbeel, and Sergey Levine. "Model-agnostic meta-learning for fast adaptation of deep networks." In International conference on machine learning, pp. 1126-1135. PMLR, 2017.

**Summary Of The Paper:**

This work focuses on graph model selection without training by matching the meta-feature and model representations. Extensive experiments are conducted to demonstrate the effective and efficiency of the proposed method. Basically, the overall idea has close relation to pioneering work MetaOD. By comparison, this work adopts the key idea to explore the graph model selection task with meta-knowledge.

**Summary Of The Review:**

This work attempts to facilitate model selection just by matching the meta-knowledge extracted from graph data and the trained models. It's a very interesting idea. But graph data raises unique challenges over the knowledge transferring across different graphs. Considering this point, it's difficult to believe the generalization of the proposed method.

---

> ### Author Response · Authors · 2022-11-19
> **Response to Reviewer uUz4**
>
> Thank you for your thoughtful comments. Please find our response to your comments and questions below. We have also updated and highlighted the paper in light of your comment, which is marked in brick red in the revised paper.
>
> > Q1a. “The novelty is limited to applying the principles of MetaOD to graph data by matching meta-features of graph data with model representations.“
>
> As noted, the proposed meta-graph features are a novel contribution that enables existing meta-learners including MetaOD applicable to GL model selection problem. In addition to the meta-graph features, we would also like to highlight that, while MetaGL is inspired by MetaOD, MetaGL makes significantly novel and effective modeling choices differently from MetaOD. First, they adopt different meta-training objectives. MetaGL adopts the top-1 probability for meta-training, while MetaOD uses an NDCG-based objective. Furthermore, MetaGL learns the embeddings of models and graphs by applying a heterogeneous GNN over the proposed G-M network, which allows a flexible modeling of the relations between different models and graphs. By contrast, in MetaOD, the embeddings of models and datasets are optimized separately, where the relations between models and datasets are modeled rather indirectly (via reconstructing the performance matrix). In experiments where MetaGL, MetaOD, and other baselines were all given the same meta-graph features, MetaGL performed more effectively than MetaOD due to these differences in modeling choices. We highlighted these differences in Appendix H.1.
>
> > Q1b. “It's interesting to compare the way to run "meta-learning" in this work with the optimization-based meta-learning methods like MAML [1]. The common thing shared by them is that finally selected model should have impressive prediction performance. The principles emphasized in this work pays attention to select the best one from the stored models which are trained on historical data. While optimization-based meta-learning can capture the unique knowledge existing in the input data just by fine-tuning models. I strongly suggest to have a discussion on the difference between them and make a comparison over their prediction performance.”
>
> MAML employs meta-learning to train a model's initial parameters such that the model can perform well on a new task after the parameters have been updated via a few gradient steps using the data from the new task. In other words, given a model, MAML initializes one specific model's parameters via meta-learning over multiple observed tasks, such that the meta-trained model can quickly adapt to a new task after learning from a small number of new data (i.e., few-shot learning). On the other hand, MetaGL employs meta-learning to carry over the prior knowledge of multiple different models' performance on different graphs for evaluation-free selection of graph learning algorithms.
>
> Since MAML meta-trains a specific model for fast adaptation to a new dataset, it is not for selecting a model from a model set consisting of a wide variety of learning algorithms. Further, MAML fine-tunes a meta-trained model in a few-shot learning setup, whereas in our problem setup, no training and evaluation is to be done given a new graph dataset. Due to these reasons, MAML is not applicable to the proposed evaluation-free GL model selection problem (Problem 1 in Section 2). We added a comparison with MAML in Appendix H.2.
>
> > Q2. “Though the proposed method can fast search one model from the candidates, it's still difficult to believe that only comparing the meta-features extracted from graph structures can select the exact model for the input data. What if the extracted meta-feature can not represent the node features distribution differences existing in the graphs? Is it still the best choice for model selection? or is the learning-free model selection suitable to deal with the knowledge gap between the new graph and those graphs in the data pool?”
>
> MetaGL transfers relevant prior experience (i.e., observed model performances on different graphs) to estimate how well those graph learning models would perform on the new graph. When all of the graphs in the benchmark dataset are dissimilar from the new graph, there isn’t much knowledge that can be transferred, and accordingly, it would be difficult for MetaGL, as well as any other methods that operate based on the same assumption, to accurately estimate model performances on the new graph.
>
> At the same time, the current benchmark graphs already cover 20+ different domains (such as information networks, social networks, road networks, biology networks, among others), and as the benchmark dataset continues to grow and cover more types of graphs, it will become less likely to encounter a new graph completely different from the ones in the graph data pool.

---

> > ### Author Response · Authors · 2022-11-19
> > **Response to Reviewer uUz4 (Cont.)**
> >
> > > Q3. “I still have concern on the generalization of the proposed method. It's well known that graph consists of unique node sets. How is it possible that the models trained on different graphs can be applied to a graph with totally different nodes? Maybe authors should present some concrete applications that are suitable to be applied to the proposed method.”
> >
> > To be able to compare different graphs that have different sets of nodes, we adopt a two-stage approach, which transforms a graph into meta-graph features of a fixed size (that can be computed on any arbitrary graph as they represent functions over a graph). The proposed meta-graph features are designed to capture the characteristics of a graph at node-, edge-, and graph-levels. Since two different graphs have different sets of nodes and edges, their node- or edge-level features obtained by the first step of our feature extraction process (described in Sec 4.3) cannot be compared directly. MetaGL makes them comparable by applying global statistical feature extractors to the distributions obtained by the first step (e.g., node degree distribution). These global statistical feature extractors (e.g., mean, skewness and correlation; Table 5 in Appendix D lists those functions) summarize graph characteristics from multiple statistical viewpoints. Since the same set of global statistical functions are applied to each graph, the resulting meta-graph features all have the same size and semantics, and thus can be used for measuring similarities between disparate graphs.
> >
> > Also, our experiments are designed to test the generalizability of model selection frameworks when they are deployed in practice. For instance, you may have a new graph that has not been used before, and now want to choose which link prediction model to use for the new graph. To reflect situations like this, we split the benchmark graphs into meta-train and meta-test graphs that do not overlap, and train MetaGL using the train graphs alone. Since no information about the test graphs was involved in the training of MetaGL and baselines, the reported experimental results represent the generalization capability of different model selection methods, in which MetaGL consistently outperforms existing approaches.

---

### Official Review · Reviewer_GdF8 · 2022-10-30

**Confidence:** 4
**Correctness:** 4
**Technical Novelty And Significance:** 3
**Empirical Novelty And Significance:** 3
**Recommendation:** 8

**Clarity, Quality, Novelty And Reproducibility:**

- The manuscript is well-written so that readers can easily follow.
- Through analysis has been made so the impact by the proposed meta-learning can be well understood.
- The idea of using graph structure as the meta-learner features is novel enough. It is worth being studied more in the research community.
- Without the shared performance matrix, it may not be trivial to reproduce the results. For the full reproducibility, all the hyperparameters chosen for performance matrix need to be shared.

**Strength And Weaknesses:**

* Strength
- The proposed meta-learning estimates the performance and leverages the predictive power for more robustly good model learning.
- The proposed method is a novel way of ensembling graph learning algorithms.
- The small initial overhead results in more reliable performance.

* Weakness
- The model performance matrix P is a random matrix since the performance is not deterministic. It would be great to study the sensitivity to the variance of the performance matrix P, not just the observation rate.
- It would be great if the proposed meta-learning is not only used for the model selection, but also for the insights about the performance characteristics of each graph model algorithm -- such as some algorithm working better for a homophily networks or clustering the similar flavor of algorithms, and so on.
- It would be interesting to compare the performance from MetaGL with the actual best performance. Since we have a fixed set of algorithms and perform training for each method, we are able to compare with the "optimal" selection.

**Summary Of The Paper:**

Authors provide the meta-learning framework for graph learning models that estimates the model performance based on the meta-graph features such as network structures. The proposed method optimizes the numerical performance values based on the top-1 probability based cross-entry loss function. Experimental results show that the proposed method can be robustly optimal for various datasets.

**Summary Of The Review:**

While GNN becomes popular in the ML area, it often requires a lot of resource to train on large datasets and chooses the best working model. The proposed meta-learning will provide the opportunity to produce more reliable results in a more productive fashion. Also, the thorough study for the given MetaGL provides understanding and confidence of the proposed model.

---

> ### Author Response · Authors · 2022-11-19
> **Response to Reviewer GdF8**
>
> Thank you for your thoughtful comments. Please find our response to your comments and suggestions below. We have also updated and highlighted the paper in light of your comment, which is marked in orange color in the revised paper.
>
> > Q1. “The model performance matrix P is a random matrix since the performance is not deterministic. It would be great to study the sensitivity to the variance of the performance matrix P, not just the observation rate.”
>
> Thank you for the suggestion. We have added an experiment in Appendix I.4 to evaluate the sensitivity of MetaGL and baseline methods to the variance of the performance matrix P. For evaluation, we perturb all entries of P to varying degrees, and measure the model selection performance using the perturbed performance matrix P’. Specifically, given a performance perturbation rate r, we replace each performance entry n in P with the perturbed performance n’, which is uniformly randomly sampled from [n(1 − r /2), n(1 + r /2)], bounded by 0 and 1. For instance, given a perturbation rate of 0.2, a mean average precision of 0.5 is perturbed by sampling from [0.45, 0.55]. As a result, a higher perturbation rate makes the perturbed matrix more random, and thus harder to predict. Figure 11 shows that MetaGL consistently outperforms baselines across varying performance perturbation rates, obtaining up to 53% higher MRR than the best baseline.
>
> > Q2. “It would be great if the proposed meta-learning is not only used for the model selection, but also for the insights about the performance characteristics of each graph model algorithm -- such as some algorithm working better for a homophily networks > or clustering the similar flavor of algorithms, and so on.”
>
> We appreciate your insightful comment. We think that the suggested idea can be used not only for identifying the performance characteristics of graph learning algorithms, but also for enabling MetaGL to perform more effective meta-learning using those characteristics. Currently, the performance matrix is a major source of information for estimating the similarities between GL algorithms. As suggested, additional knowledge that we have about the GL algorithm can be incorporated, e.g., explicitly modeling the semantics of algorithm configurations such that two methods whose configurations are close would be considered similar. These extensions may make MetaGL to be more effective, especially when the performance matrix is sparse. We will take these suggestions into account when we extend MetaGL.
>
> > Q3. “It would be interesting to compare the performance from MetaGL with the actual best performance. Since we have a fixed set of algorithms and perform training for each method, we are able to compare with the "optimal" selection.”
>
> Following this suggestion, we compared the actual best performance observed for each graph against the performance of the model chosen by different model selection approaches (Appendix I.5 and Figure 12). Figure 12 shows that using MetaGL leads to the smallest performance difference between the actual best and the predicted best models (27% and 20% and less than the second smallest difference obtained by ALORS in terms of median and mean, respectively). Further, the performance difference distribution obtained with MetaGL is much more tight than that of baselines.
>
> [Reproducibility] We will make the code and benchmark datasets publicly available when the paper gets published.

---

### Decision · Program_Chairs · 2023-01-20

**Decision:**

Accept: poster

**Justification For Why Not Higher Score:**

The paper focuses on model selection for link prediction. However, the tested models exclude some of the most well-known link prediction baselines. Without appropriate baselines, it is hard to verify whether the proposed meta-learning framework can actually pick the best model for each graph.

**Justification For Why Not Lower Score:**

N/A

**Metareview: Summary, Strengths And Weaknesses:**

This paper proposes a meta-learning framework for evaluation-free model selection of graph link prediction models. The paper proposes a new problem where given a performance matrix of M models on N graphs, and a new unseen graph, select a model among M that performs best on the new graph (without training/evaluation). This becomes a cold-start matrix completion problem with a new column added. Structural meta-graph features, matrix factorization, and graph neural network inference on model-graph bipartite grap are used. The paper uses link prediction as the target graph learning task. Experiments on 301 graphs with 412 candidate models verify the effectiveness of the proposed framework.

I am recommending an accept for this paper, but strongly suggest the authors to address the following problems in their revision:

1. Although the proposed problem is new and interesting, the authors should better explain what the problem's real applications could be. That is, find a senario where people need to repeatedly train models on different graphs and generalize them to new graphs in a evaluation-free setting. As far as I know, industry applications highly focus on model accuracy as it is tied with revenue or user experience. It may be hard to trust a meta-learned strategy for model selection.

2. Holding similar concerns to Reviewer uUz4, without modeling node features, it is questionable whether only comparing the meta-features extracted from graph structures can select the best model for the input graph. Also, there are many transductive network embedding models such as node2vec/LINE in the model pool. The authors may need better explain how one can generalize node embeddings trained in one graph to another graph (with a different set of nodes) without re-training.

3. The authors claim choosing link prediction as a testbed because it is a key task for graph data. However, the link prediction baselines used are mostly outdated or are not specifically targeted for link prediction (known to perform poor on link prediction). The authors should include more modern link prediction baselines, such as SEAL and NBFNet, in their model pool. Without appropriate baselines, it is hard to verify the true effectiveness of the proposed framework.

**Note From Pc:**

if the above contains the word "oral" or "spotlight" please see: "oral" presentation means -> notable-top-5% and "spotlight" means -> notable-top-25%. As stated in our emails, we are disassociating presentation type from AC recommendations